# Spreading of a mycobacterial cell-surface lipid into host epithelial membranes promotes infectivity

CJ Cambier[1], Steven M Banik[1], Joseph A Buonomo[1], Carolyn R Bertozzi[1,2]*

[1]Department of Chemistry, Stanford University, Stanford, United States; [2]Howard Hughes Medical Institute, Stanford University, Stanford, United States

**Abstract** Several virulence lipids populate the outer cell wall of pathogenic mycobacteria. Phthiocerol dimycocerosate (PDIM), one of the most abundant outer membrane lipids, plays important roles in both defending against host antimicrobial programs and in evading these programs altogether. Immediately following infection, mycobacteria rely on PDIM to evade Myd88-dependent recruitment of microbicidal monocytes which can clear infection. To circumvent the limitations in using genetics to understand virulence lipids, we developed a chemical approach to track PDIM during *Mycobacterium marinum* infection of zebrafish. We found that PDIM's methyl-branched lipid tails enabled it to spread into host epithelial membranes to prevent immune activation. Additionally, PDIM's affinity for cholesterol promoted this phenotype; treatment of zebrafish with statins, cholesterol synthesis inhibitors, decreased spreading and provided protection from infection. This work establishes that interactions between host and pathogen lipids influence mycobacterial infectivity and suggests the use of statins as tuberculosis preventive therapy by inhibiting PDIM spread.

## Introduction

*Mycobacterium tuberculosis*, the causative pathogen of the pulmonary disease tuberculosis (TB), is estimated to have evolved within the confines of the human lung for millennia (*Comas et al., 2013*). A result of this co-evolution is a choreographed response of innate and adaptive immune cells culminating in the formation of granulomas, specialized structures that permit bacterial replication and ultimately promote transmission (*Ramakrishnan, 2012*). A key strategy used by mycobacteria throughout infection is to avoid and manipulate host immune pathways so as to afford the pathogen safe harbor in otherwise bactericidal myeloid cells (*Cambier et al., 2014a*; *Urdahl, 2014*).

To better understand these host–pathogen interactions, we have taken advantage of the optically transparent zebrafish larva, a natural host of the pathogen *M. marinum* (*Ramakrishnan, 2020*; *Takaki et al., 2013*). Infection of the hindbrain ventricle (HBV), an epithelium-lined cavity, allows for the visualization and characterization of the cellular immune response (*Davis and Ramakrishnan, 2009*), a response that is comparable to that seen in the mouse lung following infection with *M. tuberculosis* (*Srivastava et al., 2014*). In both models, mycobacteria are initially phagocytosed by tissue-resident macrophages and are eventually transferred to monocytes which go on to form granulomas (*Cambier et al., 2017*; *Cohen et al., 2018*).

In order to reach growth-permissive cells, mycobacteria must first evade prototypical anti-bacterial monocytes. In response to mucosal commensal pathogens, bactericidal monocytes are recruited downstream of toll-like receptor (TLR) signaling (*Medzhitov, 2007*). Screening of *M. marinum* genetic mutants found that the cell-surface lipid phthiocerol dimycocerosate (PDIM) is required to evade this antibacterial response (*Cambier et al., 2014b*). PDIM also promotes pathogenesis in other ways, such as being required for the relative impermeability of the mycobacterial cell wall

*For correspondence:
bertozzi@stanford.edu

(*Camacho et al., 2001*) and in promoting escape from phagolysosomes (*Augenstreich et al., 2017*; *Barczak et al., 2017*; *Lerner et al., 2017*; *Quigley et al., 2017*). However, the molecular details underlying PDIM's myriad pathogenic functions remain unknown.

To accomplish mechanistic studies of virulence lipids, we and others developed metabolic labeling strategies where unnatural metabolic precursors are fed to growing bacteria (*Siegrist et al., 2015*). The unnatural metabolite contains a bioorthogonal functional group that facilitates visualization of macromolecules in living bacteria (*Sletten and Bertozzi, 2009*). An example is the labeling of trehalose containing lipids with azide-functionalized trehalose (*Swarts et al., 2012*).

Here, we have developed comparable chemical tools to monitor PDIM's distribution during infection. We found that the first step in PDIM-mediated pathogenesis is to spread into epithelial cells in order to prevent the recruitment of microbicidal monocytes. Structure function analysis revealed that PDIM's methyl-branched fatty acids increased lipid mobility and promoted spread. Spreading was also dependent on the lipid content of host membranes. Administration of the cholesterol lowering drug, atorvastatin (Lipitor), led to a decrease in PDIM spreading, and subsequent resistance to mycobacterial infection. Our findings provide a mechanistic explanation for the association of statin use with a decrease in TB incidence (*Lai et al., 2016*) and support their use as a TB preventative therapy.

## Results

### Lipid removal and recoating of *M. marinum*

PDIM lacks unique biosynthetic precursors to facilitate metabolic labeling (*Onwueme et al., 2005*). However, PDIM is removed following petroleum ether extraction (*Moliva et al., 2019*), a technique used to remove and add back mycomembrane lipids (*Silva et al., 1985*). Using this approach, we hypothesized that we could chemically install a biorthogonal handle onto extracted PDIM and use this modified lipid to elucidate the fundamental mechanisms underlying PDIM's contribution to virulence. Similar to reports on *M. tuberculosis* and *M. bovis,* we validated that petroleum ether extraction did not affect the growth of *M. marinum* in culture (*Figure 1—figure supplement 1A*) and extracted lipids did not repopulate the mycomembrane following the first few days in culture (*Figure 1—figure supplement 1B*; *Indrigo et al., 2003*). Thus, this approach is well suited for loss of function studies of outer mycomembrane lipids in zebrafish. Extracted lipids could also be mixed with bacteria in petroleum ether followed by drying to recoat the bacterial surface (*Figure 1A* and *Figure 1—figure supplement 1C*). Evaluation by thin-layer chromatography and NMR demonstrated that the lipid composition of recoated bacteria was comparable to untreated bacteria (*Figure 1—figure supplement 1D* and Materials and methods). No protein was detected in the extracts (*Figure 1—figure supplement 1E*) suggesting minimal disruption of cell wall proteins. Following infection of zebrafish (*Figure 1B*) delipidated bacteria were attenuated for growth and this phenotype was rescued upon recoating (*Figure 1C and D*).

### Pre-infection PDIM reservoirs are required for virulence

*M. marinum* mutants in PDIM synthesis (Δ*mas*) and localization to the mycomembrane (Δ*mmpL7*) trigger TLR/Myd88-dependent immune responses (*Cambier et al., 2014b*). Myd88 signaling leads to the recruitment of activated monocytes that can clear bacteria in an inducible nitric oxide synthase-dependent fashion. PDIM-sufficient wildtype bacteria do not elicit this response and instead recruit a comparable number of permissive monocytes downstream of the chemokine CCL2 (*Cambier et al., 2014b*). However, since Δ*mas* and Δ*mmpL7 M. marinum* lack proteins required for PDIM's synthesis or export, the associated phenotypes could be attributed to the missing proteins rather than to a lack of PDIM. While both of these mutants also lack the closely related phenolic glycolipid (*Onwueme et al., 2005*), results evaluating strains lacking only phenolic glycolipid ruled out this lipids role in mediating evasion of TLRs (*Cambier et al., 2014b*).

To test if the lipid content on the bacterial surface is responsible for these phenotypes, we performed a lipid-swap experiment. Wildtype and Δ*mmpL7 M. marinum* were either untreated (control) or extracted and recoated with their native lipids or the lipids from the other strain (*Figure 1E*). Petroleum ether extraction of wildtype bacteria removed both dimycocerosic acid (DIM) containing lipids, PDIM, and its metabolic precursor phthiodiolone dimycocerosate (PNDIM, *Figure 1—figure*

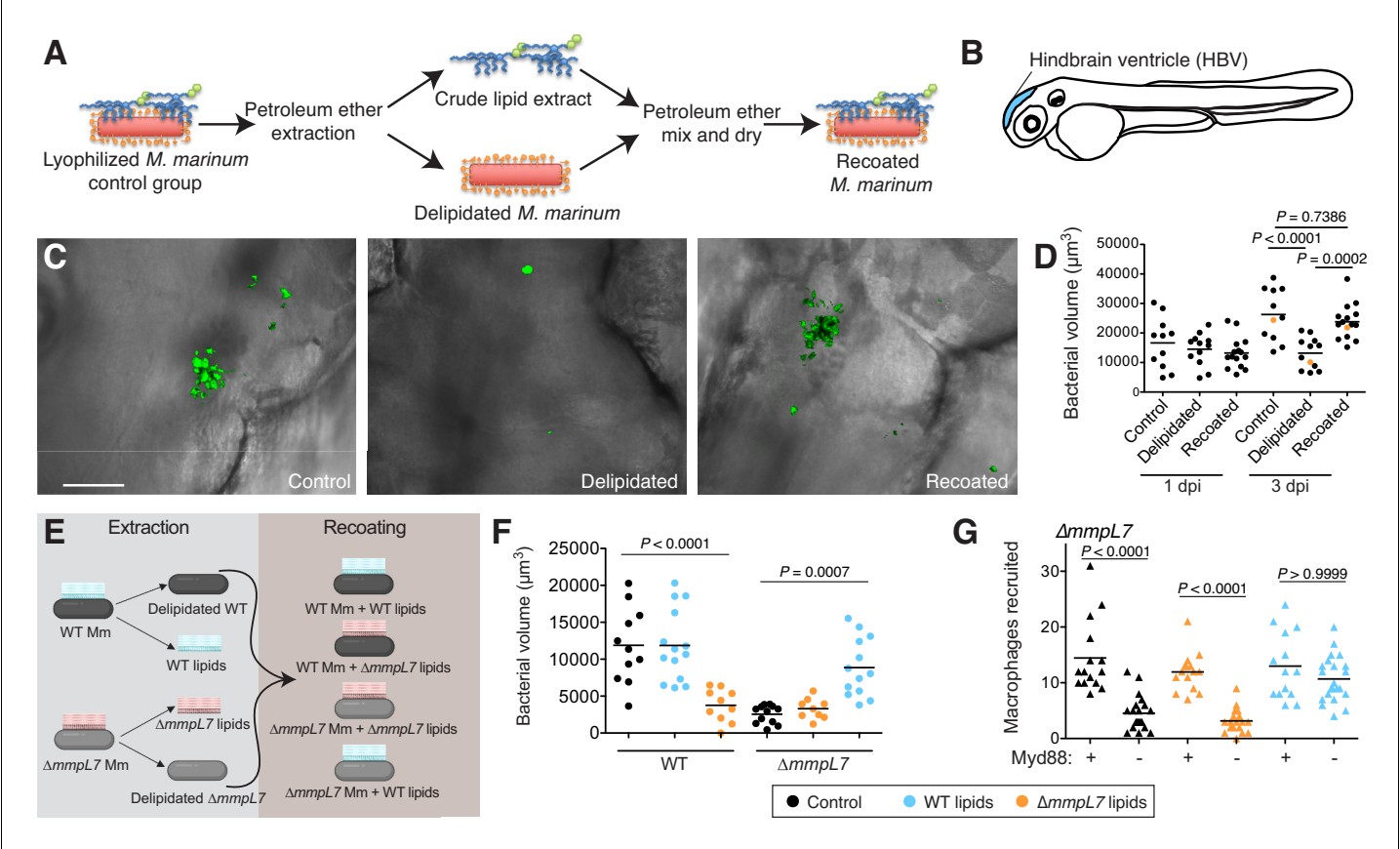

**Figure 1.** Lipid removal and recoating reveals that pre-infection PDIM reservoirs are required for *M.marinum* infection of zebrafish. (A) Model of lipid removal and recoating of *M. marinum*. (B) Model of zebrafish larva showing the hindbrain ventricle (HBV) injection site. (C) Representative images of the experiment in D (orange dots), wasabi (green) fluorescent protein expressing *M. marinum* in the HBV at 3 dpi are shown, scale bar = 50 μm. (D) Mean bacterial volume after HBV infection of wildtype fish with ~100 control, delipidated, or recoated *M. marinum*. (E) Model of lipid-swap experiment. (F) Mean bacterial volume at 3 dpi after HBV infection of wildtype fish with ~100 WT or Δ*mmpL7 M. marinum* treated as follows: non-extracted control (black), extracted and recoated with WT lipids (blue), or extracted and recoated with Δ*mmpL7* lipids (orange). (G) Mean macrophage recruitment at 3 hpi of the HBV of wildtype or Myd88-depleted fish with ~100 Δ*mmpL7 M. marinum* as treated in F. (D), (F), and (G) representative of at least three separate experiments. Ordinary one-way ANOVA with (D) Sidak's multiple comparisons test for the comparison's shown and (F) Tukey's multiple comparisons test with selected adjusted *P* values shown. (G) Kruskal-Wallis ANOVA for unequal variances with Dunn's multiple comparisons test with selected adjusted *P* values shown.

The online version of this article includes the following source data and figure supplement(s) for figure 1:

**Source data 1.**

**Figure supplement 1.** Optimization of petroleum ether extraction and recoating.

**Figure supplement 1—source data 1.**

*supplement 1F*) both of which were absent in Δ*mmpL7* extracts (*Figure 1—figure supplement 1G*). Following infection, wildtype control and wildtype bacteria recoated with wildtype lipids grew normally whereas wildtype bacteria recoated with Δ*mmpL7* lipids were attenuated for growth (*Figure 1F*). Conversely, Δ*mmpL7* bacteria were attenuated for growth, as expected, unless they were recoated with wildtype lipids, in which case they grew at wildtype bacterial rates (*Figure 1F*). Using an antisense morpholino to knockdown Myd88 (*Bates et al., 2007*), we also found that the dependence on Myd88 to recruit monocytes to Δ*mmpL7* bacteria was abolished with wildtype lipids (*Figure 1G*). Taken together these experiments highlight the strengths of this chemical approach. Not only does it recapitulate known phenotypes of PDIM genetic mutants, but it directly links the mutant phenotypes to the mycomembrane composition. Furthermore, our data suggest that the PDIM present on the surface of the bacterium from the onset of infection is required and sufficient

to promote virulence, as mutants unable to replenish PDIM on their surfaces become infectious when they are recoated with wildtype lipids.

## Synthesis of a clickable, biologically active PDIM

Given the pathogenic importance of the pre-infection mycomembrane lipid content, we hypothesized that labeling this pool of PDIM would shed light on its virulence mechanisms. Both PDIM and its biosynthetic precursor PNDIM are present in the mycomembrane. The only difference between these lipids is their diol backbones; PDIM has a methyl ether, while PNDIM has a ketone (*Siméone et al., 2007*). Either lipid can promote infection in mice (*Siméone et al., 2007*), suggesting chemical flexibility at this site with regards to virulence. Therefore, we converted the methyl ether of PDIM to an alkyl halide with trimethylsilyl iodide (*Jung and Lyster, 1977*). Subsequent addition of sodium azide provided azido-DIM (*Figure 2A*). Recoating of delipidated bacteria with lipids containing azido-DIM, followed by a copper-free click reaction with the cyclooctyne fluorophore DIBO-488 (*Figure 2B*) resulted in a ~100-fold increase in fluorescence (*Figure 2C*). Confocal microscopy revealed the fluorescence to be membrane-associated (*Figure 2D and E*), suggesting incorporation into the mycomembrane. Importantly, we found that adding back native DIMs or azido-DIM to DIM-depleted lipids prior to recoating and labeling (*Figure 2—figure supplement 1*) rescued DIM-depleted bacteria's growth attenuation (*Figure 2F*). Thus, with this approach we can generate bacteria with chemically functionalized PDIM that retain their pathogenicity.

## PDIM spreads into macrophage membranes

To visualize PDIM's distribution, we infected zebrafish with blue-fluorescent *M. marinum* that were recoated with azido-DIM followed by labeling with DIBO-488 (DIM-488). DIM-488 spread away from bacteria into infected macrophage membranes (*Figure 3—figure supplement 1A*). Real-time imaging revealed that the spreading was dynamic in nature, with DIM-488 moving relative to host cells (*Video 1*). To better visualize spreading of PDIM into macrophage membranes, we used the transgenic zebrafish line *Tg(mfap4:tdTomato)* whose macrophages express the fluorescent protein tdTomato (*Walton et al., 2015*). As early as 3 hr post-infection (hpi), DIM-488 had spread into infected macrophage membranes, directly adjacent to infecting bacteria (*Figure 3A*, arrows) and at more distal membrane sites (*Figure 3A*, arrow heads). These data suggest that lateral diffusion as well as propagation into discrete membrane compartments by PDIM is taking place. Spreading increased across macrophage membranes by 3 days post-infection (dpi) (*Figure 3B*). Similar spreading was seen when azido-DIM was conjugated to DIBO-647 (*Figure 3—figure supplement 1B and C*), suggesting that the lipid, not the fluorescent probe, was responsible for this phenotype. To quantify the extent of PDIM spreading, we imaged the entire HBV infection site and calculated the proportion of fluorophore labeled azido-DIM that no longer localized with bacteria (*Figure 3—figure supplement 1D*). Using this number as a proxy for lipid spread, we saw an increase in spreading as infection progressed (*Figure 3C*). Spreading also occurred following infection of THP-1 macrophages in culture (*Figure 3—figure supplement 1E*). Finally, PDIM spreading was not a result of homeostatic lipid turnover. DIM-488 labeled bacteria were fluorescent following 3 days in culture (*Figure 3D*), and signal remained localized to the cell wall (*Figure 3E*). Thus, PDIM spreading away from bacteria only occurs following interactions with host cells.

These results were consistent with a recent report that *M. tuberculosis* PDIM occupies cultured macrophage membranes (*Augenstreich et al., 2019*). Nevertheless, we wanted to rule out spreading as an artifact of our recoating method. We used an established pan-glycolipid labeling method previously used to track mycobacterial glycolipids through macrophage membranes (*Beatty et al., 2000*). Control and recoated *M. marinum* were treated with periodate and then reacted with a fluorescent hydroxylamine prior to infection (*Beatty et al., 2000*). We found equal spreading of the total pool of fluorophore-labeled glycolipids (*Figure 3F* and *Figure 3—figure supplement 1F*). Thus, recoating does not appreciably influence the spreading dynamics of mycomembrane lipids. These data demonstrate that the introduction of a chemically functionalized PDIM into the mycomembrane provides relevant information regarding PDIM's host distribution during infection.

We next wanted to understand how PDIM spreading might be promoting virulence. PDIM has been suggested to interact with the protein substrates of the type VII secretion system ESX-1, including EsxA (*Barczak et al., 2017*). PDIM and EsxA are both required for cytosolic escape from

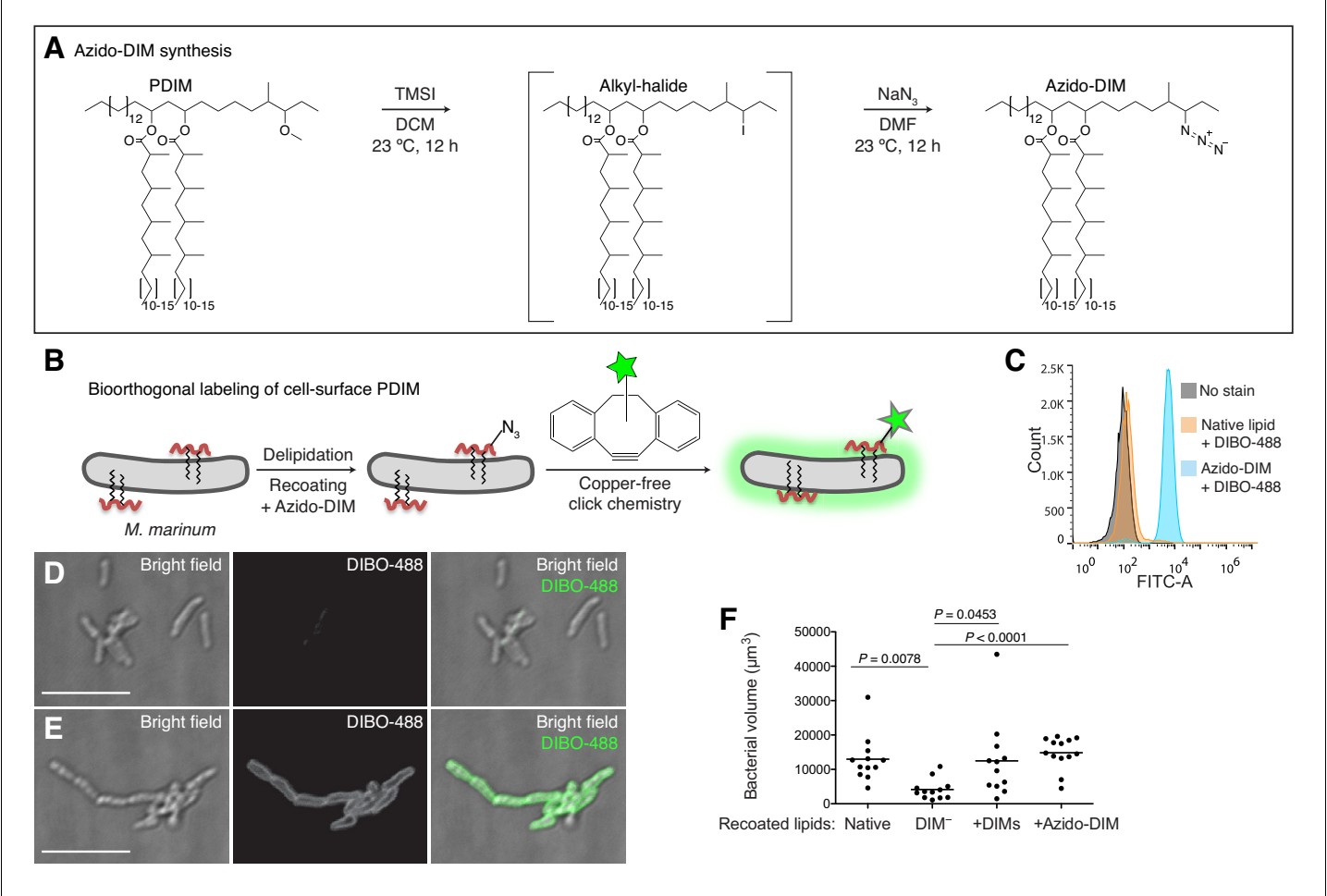

**Figure 2.** Synthesis and application of a chemically tractable, biologically active PDIM variant, azido-DIM. (A) Synthesis of azido-DIM. (B) Model of delipidation and recoating of bacteria with or without azido-DIM followed by treatment with an azide-reactive cyclooctyne, DIBO-488. (C) Flow cytometry analysis of *M. marinum* recoated with or without azido-DIM treated with DIBO-488. Image of (D) native lipid control or (E) azido-DIM recoated bacteria treated with DIBO-488, scale bar = 8 μm. (F) Mean bacterial volume 3 days following HBV infection of wildtype fish with ~100 delipidated *M. marinum* recoated with Native, DIM-depleted (DIM⁻), DIM⁻ plus native DIMs (+DIMs), or DIM⁻ plus azido-DIM (+Azido-DIM) lipids. Kruskal-Wallis ANOVA for unequal variances with Dunn's multiple comparisons test with selected adjusted *P* values shown. (C), (F) representative of three separate experiments.

The online version of this article includes the following source data and figure supplement(s) for figure 2:

**Source data 1.**

**Figure supplement 1.** Thin-layer chromatography of Native, DIM-depleted (DIM⁻), DIM⁻ plus native DIMs (+DIMs), or DIM⁻ plus azido-DIM (+Azido-DIM) lipids prior to recoating onto delipidated bacteria.

phagolysosomes (*Osman et al., 2020*; *Quigley et al., 2017*; *van der Wel et al., 2007*), where PDIM is suggested to enhance the pore-forming activity of EsxA through its ability to infiltrate macrophage membranes (*Augenstreich et al., 2017*). Thus, we hypothesized that PDIM's localization may be dependent on EsxA's pore forming ability. Region of difference-1 *M. marinum* mutants (ΔRD1) which lack EsxA (*Volkman et al., 2010*), were recoated with DIM-488 prior to zebrafish infection. There was no difference in DIM-488 spreading kinetics between wildtype and ΔRD1 *M. marinum* (*Figure 3G*), suggesting that EsxA does not influence PDIM spreading. Besides playing a role in cytosolic escape, PDIM has also been shown to promote phagocytosis of extracellular bacteria (*Astarie-Dequeker et al., 2009*; *Augenstreich et al., 2019*). To determine the phagocytosis rate of wildtype or Δ*mmpL7 M. marinum* in vivo, we measured the number of discrete bacterial objects over time. As bacteria are phagocytosed by macrophages, individual bacteria can no longer be

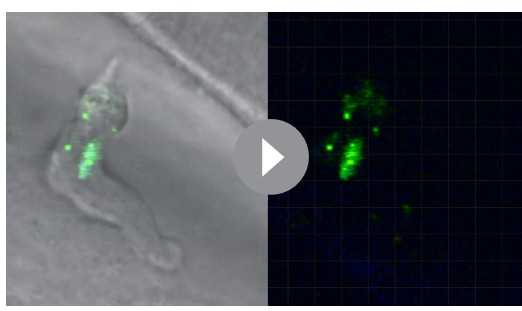

**Video 1.** PDIM dynamics. Real-time video of *M. marinum* expressing blue-fluorescent protein recoated with DIBO-488 labeled azido-DIM at 3 hpi of the HBV with ~100 *M. marinum*.
https://elifesciences.org/articles/60648#video1

discerned by confocal microscopy and the number of objects decreases. There was no measurable difference in the rate of phagocytosis of wildtype or Δ*mmpL7* bacteria (*Figure 3H*).

## PDIM spreads into epithelial membranes

Given the discrepancy regarding PDIM's role in promoting phagocytosis between the cultured macrophage and zebrafish models, we wondered if the activation state of responding immune cells in zebrafish larvae was influencing their phagocytic capacities. One clue to the timing of PDIM's role in virulence was the kinetics of the myeloid response. Wildtype bacteria needed to reside within resident macrophages in order to recruit permissive monocytes. In contrast, PDIM-deficient bacteria recruited microbicidal monocytes independent of and concurrent to resident macrophages (*Cambier et al., 2017*). Therefore, we wondered if PDIM plays a critical role in evading immune detection prior to any of its documented roles in modulating macrophages. Upon closer examination, we observed DIM-488 deposits on zebrafish epithelium at 24 hpi (*Figure 4—figure supplement 1*). Imaging at 3 hpi we captured extracellular bacteria having spread DIM-488 in the vicinity of an infected macrophage (*Figure 4A*). To better visualize spreading on these cells, we injected bacteria intravenously into the transgenic zebrafish line *Tg(flk1:mcherry)*, which has a red-fluorescent vascular endothelium (*Wang et al., 2010*). We found that the DIM-488 from bacteria contacting endothelium had spread away from the bacteria onto the surrounding tissue (*Figure 4B*, arrows). To confirm PDIM spreading into epithelial membranes, we infected human A549 epithelial cells whose plasma membranes were labeled with Alexa-fluor 594 wheat germ agglutinin. We observed DIM-488 spreading into labeled epithelial plasma membranes (*Figure 4C*, arrows).

To address the timing of PDIM spread into epithelial and macrophage membranes we imaged infected zebrafish at 2 hpi of the HBV, prior to macrophage phagocytosis, and at 4, 6, and 12 hpi by which time the majority of bacteria reside within macrophages. Preceding any appreciable spreading onto macrophages, DIM-488 had already spread onto epithelial cells by 2 hpi (*Figure 4D* yellow outlines, and *Figure 4E*). At later timepoints, where bacteria are found increasingly within macrophages (*Figure 3H*), additional spreading is seen occurring on macrophage membranes. These data suggest that mycobacteria spread PDIM onto epithelial cells prior to interactions with macrophages, and that this spread PDIM remains within these epithelial cells even after bacteria are phagocytosed by macrophages. By depleting macrophages from zebrafish larva using clodronate-loaded liposomes (*Bernut et al., 2014*), we confirmed that DIM-488 spreading still occurs in the absence of macrophages (*Figure 4F*). These data demonstrate that PDIM spreads onto epithelial cells independent of and prior to macrophage phagocytosis.

## PDIM's mobility promotes spread into epithelial cell membranes

We sought to understand PDIM's properties that facilitated its ability to spread into host membranes. It is well established that increased membrane fluidity, a feature influenced by the mobility of individual membrane lipids, promotes membrane mixing (*Howell et al., 1972*). Previous studies from our lab found that the mycomembrane of *Mycobacterium smegmatis* is relatively immobile by using fluorescence recovery after photobleaching (FRAP) of metabolically labeled trehalose monomycolate (TMM) (*Rodriguez-Rivera et al., 2017*). We confirmed this for *M. marinum* TMM. Using the metabolic label 6-azido-trehalose (*Swarts et al., 2012*) followed by reaction with DIBO-488 to track TMM (TMM-488), we found very little recovery following photobleaching of TMM-488, with only around 40% of the labeled lipids being mobile (*Figure 5A–C*, and *Figure 5—figure supplement 1A*). When we assessed PDIM, we found DIM-488 recovery to be very efficient, with a half-life

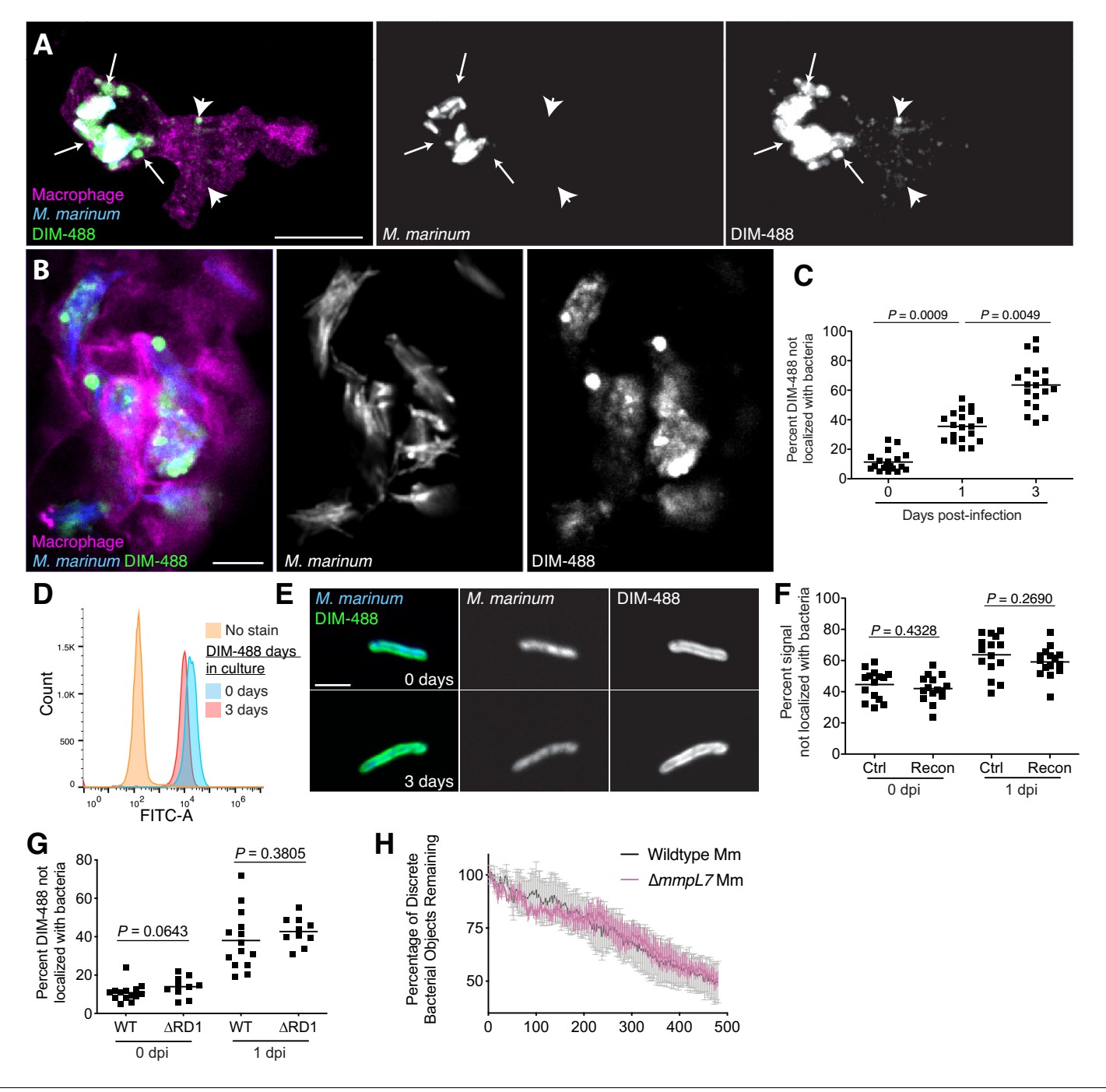

**Figure 3.** PDIM spreads into macrophage membranes. Images of *M. marinum* expressing a cytosolic blue-fluorescent protein recoated with DIBO-488 labeled azido-DIM (DIM-488) at (**A**) 3 hpi and (**B**) 3 dpi of ~100 *M. marinum* in the HBV of transgenic fish whose macrophages express a fluorescent protein. Scale bar = 10 μm. Arrows, DIM-488 spread in vicinity of infecting bacteria, arrowheads, DIM-488 spread throughout macrophage. (**C**) Mean percent DIM-488 not localized with bacteria following HBV infection of wildtype fish with ~100 *M. marinum*. Kruskal-Wallis ANOVA for unequal variances with Dunn's multiple comparisons test with selected adjusted *P* values shown. (**D**) Flow cytometry analysis of *M. marinum* expressing a cytosolic blue-fluorescent protein recoated with DIM-488 following 0 or 3 days in culture. Representative of two separate experiments. (**E**) Representative images of bacteria from D, scale bar = 3 μm. (**F**) Mean percent fluorescent signal not localized with bacteria following HBV infection of wildtype fish with ~100 control or recoated *M. marinum* labeled with periodate-hydroxylamine chemistry. Two-tailed, unpaired t test. (**G**) Mean percent DIM-488 not localized with bacteria following HBV infection of wildtype fish with ~100 wildtype or ΔRD1 *M. marinum*. Two-tailed Mann Whitney test for 0 dpi and two-tailed, unpaired t test for one dpi (**H**) Mean (+/- SEM) percentage of discrete bacterial objects remaining following HBV infection of wildtype fish with ~100 wildtype or Δ*mmpL7 M. marinum*. Representative of two separate experiments. (**C**), (**F**) and (**G**) representative of three separate experiments.

*Figure 3 continued on next page*

*Figure 3 continued*

The online version of this article includes the following source data and figure supplement(s) for figure 3:

**Source data 1.**

**Figure supplement 1.** Further characterization of PDIM spread.

of 3 s and around 95% of the signal being mobile (*Figure 5A–C*, and *Figure 5—figure supplement 1B*). Thus, PDIM lipids are more mobile than TMM lipids.

If increased membrane fluidity and lipid mobility promote membrane mixing, then TMM should not be able to spread into host cells as efficiently as PDIM does. Indeed, we found that TMM-488 failed to spread onto epithelial cells at 2 hpi (*Figure 5D*). Only when bacteria are within macrophages at 24 hpi was TMM spreading detected (*Figure 5D* and *Figure 5—figure supplement 1C*). Even in the absence of macrophages, where bacteria have a prolonged contact time with epithelial cells, TMM-488 spreading was negligible (*Figure 5E*). Thus, lipid mobility correlates with the ability to spread into host epithelial membranes.

To determine if there is a causal relationship between mobility and spreading, we used FRAP to identify conditions that decreased PDIM's recovery after photobleaching. Neither heat-killing nor mild chemical fixation (4% paraformaldehyde (PFA)) significantly reduced DIM-488's recovery (*Figure 5F,G* and *Figure 5—figure supplement 2*). However, fixation with 4% PFA + 1% glutaraldehyde (GA), a more effective fixative for membrane-associated proteins (*Huebinger et al., 2018*), resulted in an almost 80% reduction in the mobile fraction of DIM-488 (*Figure 5F,G* and *Figure 5—figure supplement 2*). PFA+GA treatment decreased DIM-488 spread onto epithelial cells at 2 hpi (*Figure 5H and I*). Likewise, fixed DIM-488 did not spread into epithelial membranes following a 24 hr infection of macrophage-depleted zebrafish (*Figure 5J*). Nor did it spread onto A549 epithelial cells after 24 hr (*Figure 5K*). Together these results suggest that PDIM's mobility promotes spreading into epithelial membranes.

## PDIM's methyl-branched mycocerosic acids promote mobility and spreading

To further test if PDIM's mobility promotes its spreading, we sought to take advantage of our ability to alter PDIM's structure. Lipid structure is known to modulate lipid mobility and membrane fluidity (*Los and Murata, 2004*). Saturated lipids form closely packed assemblies giving rise to more rigid bilayers, while unsaturated lipids do not pack as tightly and produce more fluid bilayers. Methyl branches on otherwise saturated lipids have also been shown to increase membrane fluidity (*Budin et al., 2018*; *Poger et al., 2014*). Therefore, we hypothesized that PDIM's methyl-branched mycocerosic acids enhance its mobility. To test this, we hydrolyzed PDIM with tetrabutylammonium hydroxide to isolate phthiocerol. We then esterified phthiocerol with the saturated fatty acid lignoceric acid that resulted in straight chain lipids of comparable length to mycocerosic acid. This phthiocerol di-fatty acid (PDIF) was then treated similarly to PDIM (*Figure 2A*) to yield the azide-labeled PDIF, azido-DIF (*Figure 6A*). FRAP studies confirmed that DIBO-488 labeled azido-DIF (DIF-488) was not as mobile as DIM-488, with only ~50% in the mobile fraction (*Figure 6B,C* and *Figure 6—figure supplement 1A–C*). Upon zebrafish infection, DIF-488 failed to spread onto epithelial cells at two hpi (*Figure 6D–F*). However, similar to other mycobacterial lipids (*Beatty et al., 2000*) including TMM (*Figure 5A*), DIF-488 was capable of spreading into macrophage membranes at later time-points (*Figure 6F* and *Figure 6—figure supplement 1D and E*). Thus, PDIM's mobility, mediated by its methyl-branched lipid tails, promotes its ability to spread into epithelial membranes.

## PDIM spreading into epithelial membranes is required to evade microbicidal monocytes

Is PDIM spreading into epithelial membranes required to inhibit Myd88 signaling? We first evaluated monocyte recruitment toward infecting mycobacteria to address this question. Wildtype *M. marinum* must be alive in order to stimulate resident macrophages to express CCL2 to recruit permissive monocytes (*Cambier et al., 2017*). In contrast, PDIM-deficient *M. marinum* do not depend on resident macrophages to recruit bactericidal monocytes and do so in a manner independent of bacterial viability (*Cambier et al., 2017*). We now wondered if, in order for mycobacteria to gain access to

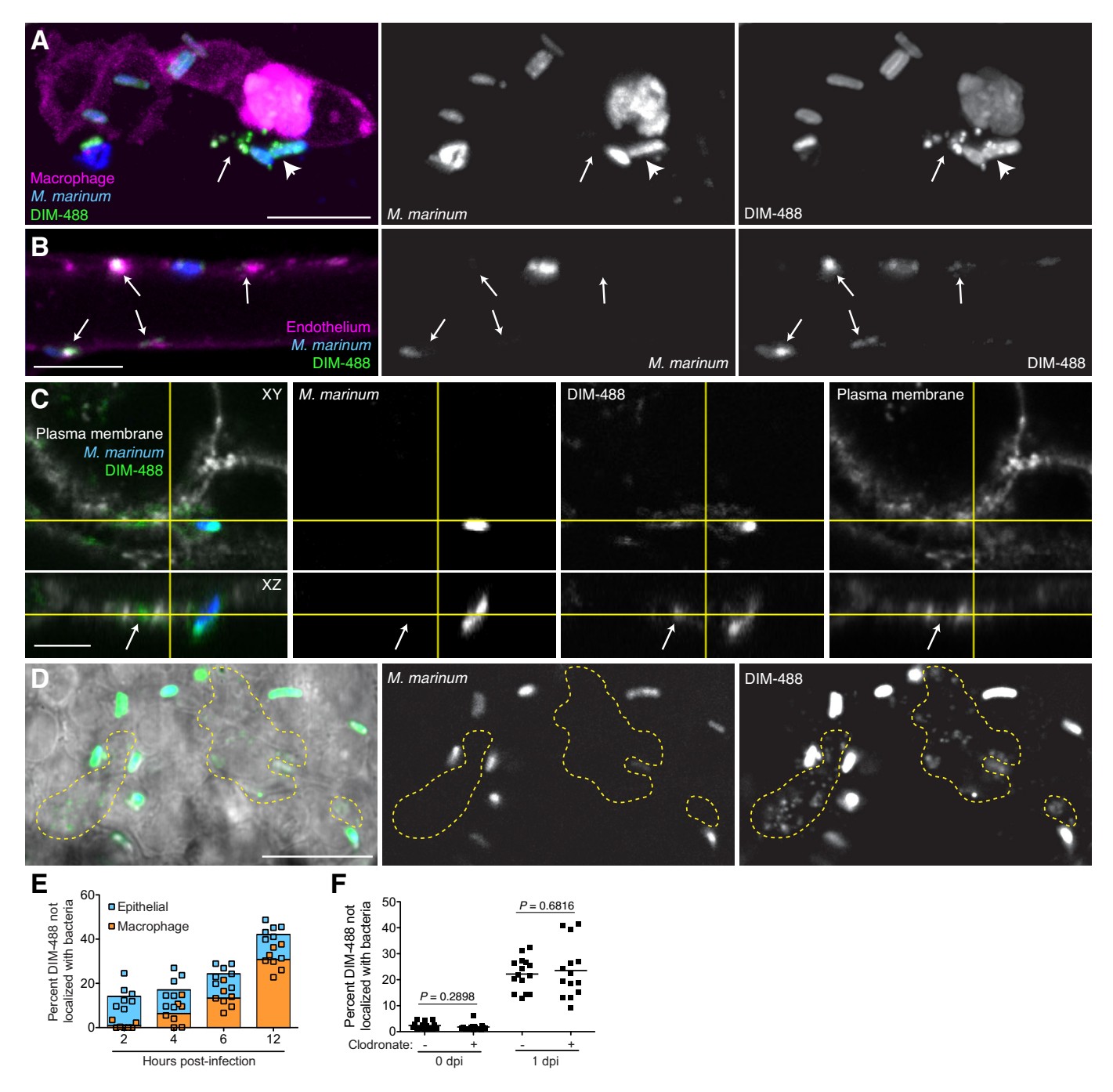

**Figure 4.** PDIM spreads into epithelial membranes. (**A**) Image of *M. marinum* expressing a cytosolic blue-fluorescent protein recoated with DIBO-488 labeled azido-DIM (DIM-488) highlighting DIM-488 spread from bacteria (arrowhead) to epithelial cells (arrows) at 3 hpi of ~100 *M. marinum* in the HBV, scale bar = 10 µm. (**B**) Image of DIM-488 labeled *M. marinum* at 1 day post-intravenous infection of transgenic fish whose endothelium express a red-fluorescent protein. Arrows, DIM-488 spread onto endothelium, scale bar = 5 µm. (**C**) Image of A549 epithelial cells whose plasma membranes are labeled with Alexa-fluor 594 wheat germ agglutinin at one day post infection with DIM-488 labeled *M. marinum* at an MOI of 5. Arrows, DIM-488 spread into plasma membrane, scale bar = 5 µm. (**D**) Image highlighting DIM-488 spread onto epithelial surfaces (yellow-dashed outline) at 2 hpi of ~100 *M. marinum* in the HBV, scale bar = 10 µm. (**E**) Mean percent DIM-488 in macrophage or epithelial cells not localized with bacteria following HBV infection with ~100 *M. marinum*. Representative of two separate experiments. (**F**) Mean percent DIM-488 not localized with bacteria following HBV infection of lipo-PBS or lipo-clodronate treated fish with ~100 *M. marinum*. Two-tailed Mann Whitney test for 0 dpi and two-tailed, unpaired t test for one dpi. Representative of three separate experiments.

The online version of this article includes the following source data and figure supplement(s) for figure 4:

*Figure 4 continued on next page*

*Figure 4 continued*

**Source data 1.**
**Figure supplement 1.** Image of *M. marinum* expressing a cytosolic blue-fluorescent protein recoated with DIBO-488 labeled azido-DIM (DIM-488) highlighting DIM-488 spread on epithelial cells (arrows) at 24 hpi of ~100 *M. marinum* in the HBV of transgenic fish whose macrophages express a fluorescent protein, scale bar = 30 μm.

resident macrophages, PDIM must first spread into epithelial membranes to prevent recruitment of microbicidal monocytes. We confirmed that heat-killed mycobacteria do not recruit monocytes (*Figure 7A*). Likewise, PFA-treated bacteria which can also spread PDIM did not recruit monocytes (*Figure 7A*). However, monocyte recruitment toward PFA+GA treated bacteria, which do not spread PDIM, phenocopied heat-killed Δ*mmpL7 M. marinum*; even though they are not viable, they

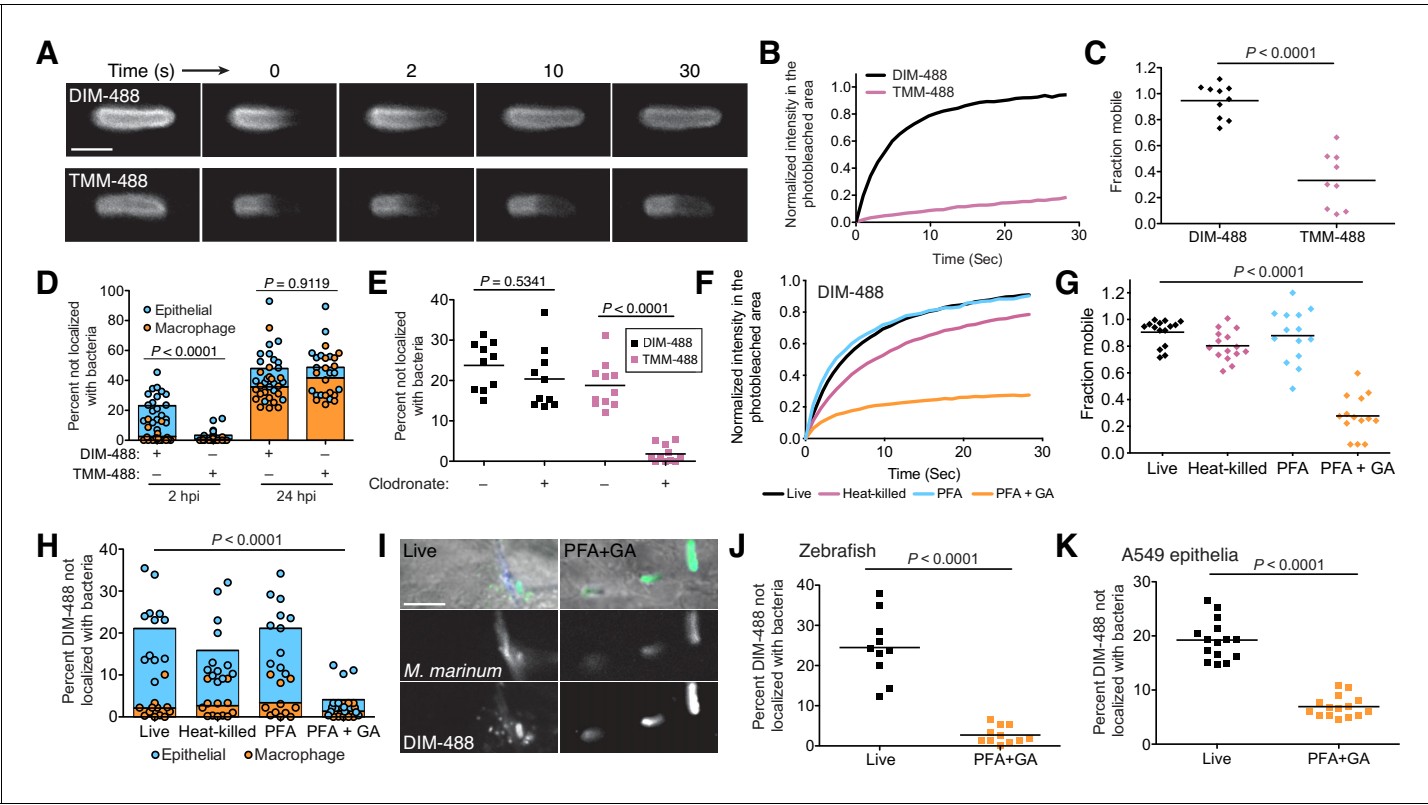

**Figure 5.** PDIM's mobility promotes spread into epithelial cell membranes. (A) Representative FRAP images of DIM-488 and TMM-488 labeled *M. marinum*, scale bar = 2 μm. (B) Fluorescent recovery curves after photobleaching of DIM-488 or TMM-488 labeled *M. marinum*, lines represent the average signal from n = 10 cells. (C) Mean fraction mobile which is the plateau following fitting of data generated in B to a non-linear regression with a one-phase association. (D) Mean percent DIM-488 or TMM-488 in macrophage or epithelial cells not localized with bacteria following HBV infection with ~100 *M. marinum*. (E) Mean percent DIM-488 or TMM-488 not localized with bacteria 24 hr following HBV infection of lipo-PBS or lipo-clodronate treated fish with ~100 *M. marinum*. (F) Fluorescent recovery curves after photobleaching of live, heat-killed, 4% paraformaldehyde (PFA) fixed, or 4% paraformaldehyde plus 1% glutaraldehyde (PFA+GA) fixed DIM-488 labeled *M. marinum*, lines represent the average signal from n = 14–15 cells. (G) Mean fraction mobile which is the plateau following fitting of data generated in F to a non-linear regression with a one-phase association. (H) Mean percent DIM-488 in macrophage or epithelial cells not localized with bacteria 2 hr following HBV infection with ~100 *M. marinum* treated as in F. (I) Images of live or PFA+GA treated DIM-488 labeled *M. marinum* at 2 hpi of the HBV with ~100 bacteria, scale bar = 5 μm. Mean percent DIM-488 not localized with bacteria 24 hr following (J) infection of lipo-clodronate treated fish or (K) A549 epithelial cells with live or PFA+GA fixed DIM-488 labeled *M. marinum*. (C), (J), and (K) two-tailed, unpaired t test. (E), (G), and (H) ordinary one-way ANOVA with Tukey's multiple comparisons test with selected adjusted *P* values shown. (B)-(H) and (J)-(K) representative of three separate experiments.

The online version of this article includes the following source data and figure supplement(s) for figure 5:

**Source data 1.**
**Figure supplement 1.** Individual FRAP curves of (A) TMM-488 and (B) DIM-488 labeled bacteria.
**Figure supplement 2.** FRAP analysis of DIM-488 labeled bacteria.

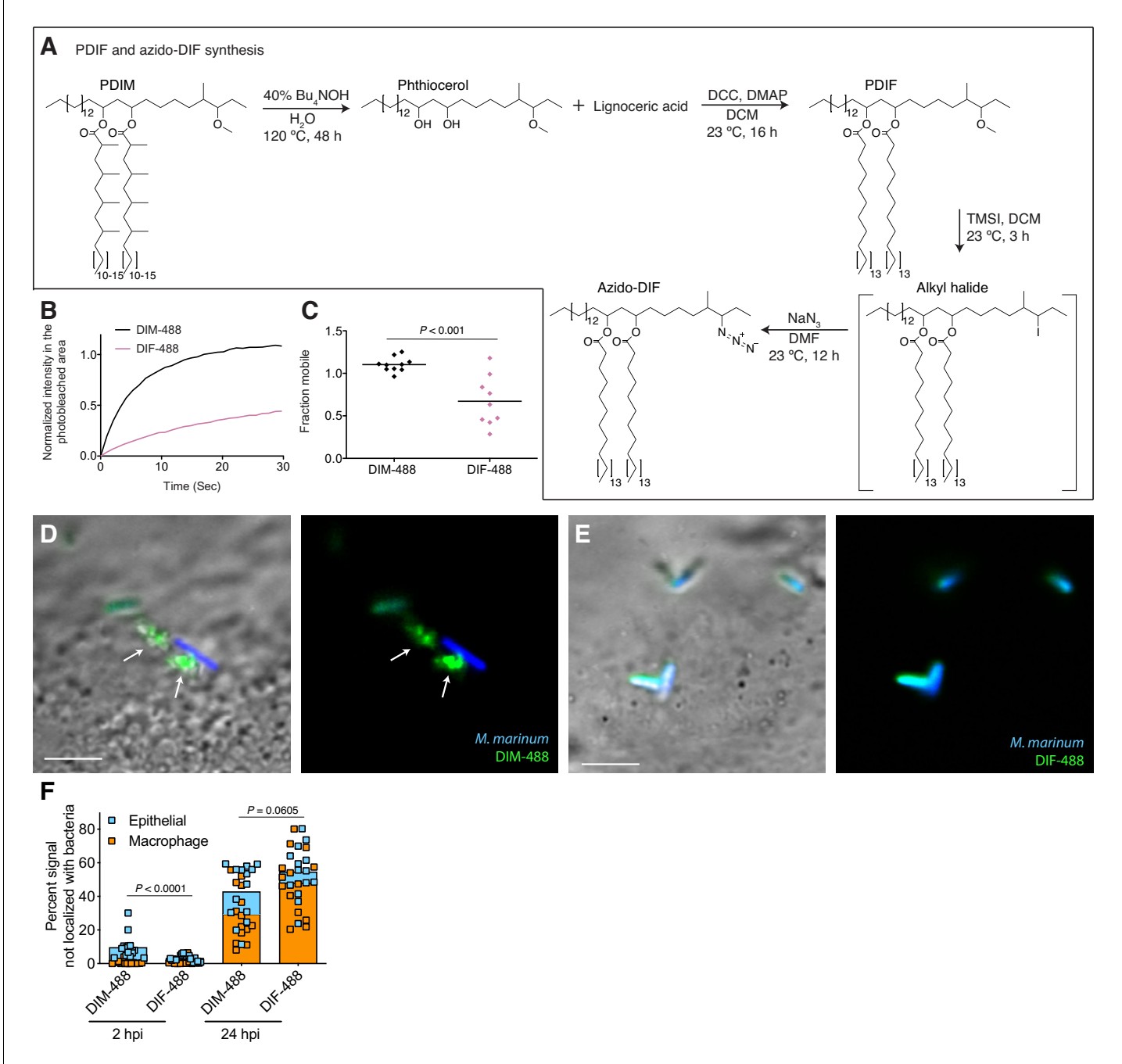

**Figure 6.** PDIM's methyl-branched mycocerosic acids promote mobility and spreading. (**A**) Phthiocerol di-fatty acid (PDIF) and azido-DIF synthesis. (**B**) Fluorescent recovery curves after photobleaching of DIM-488 or DIF-488 labeled *M. marinum*, lines represent the average signal from n = 9–10 cells. (**C**) Mean fraction mobile which is the plateau following fitting of data generated in B to a non-linear regression with a one-phase association. Two-tailed, unpaired t test. Images of *M. marinum* expressing a blue-fluorescent protein recoated with (**D**) DIM-488 or (**E**) DIF-488 at 2 hpi into the HBV of wildtype fish, arrows indicate spread signal, scale bar = 5 µm. (**F**) Mean percent DIM-488 or DIF-488 in macrophage or epithelial cells no longer localized with bacteria following HBV infection with ~100 *M. marinum*. Two-tailed Mann Whitney test for 2 hpi and two-tailed, unpaired t test for 24 hpi. (**B**), (**C**), and (**F**) representative of three separate experiments.

The online version of this article includes the following source data and figure supplement(s) for figure 6:

**Source data 1.**

**Figure supplement 1.** Analysis of DIM-488 and DIF-488 labeled *M.marinum*.

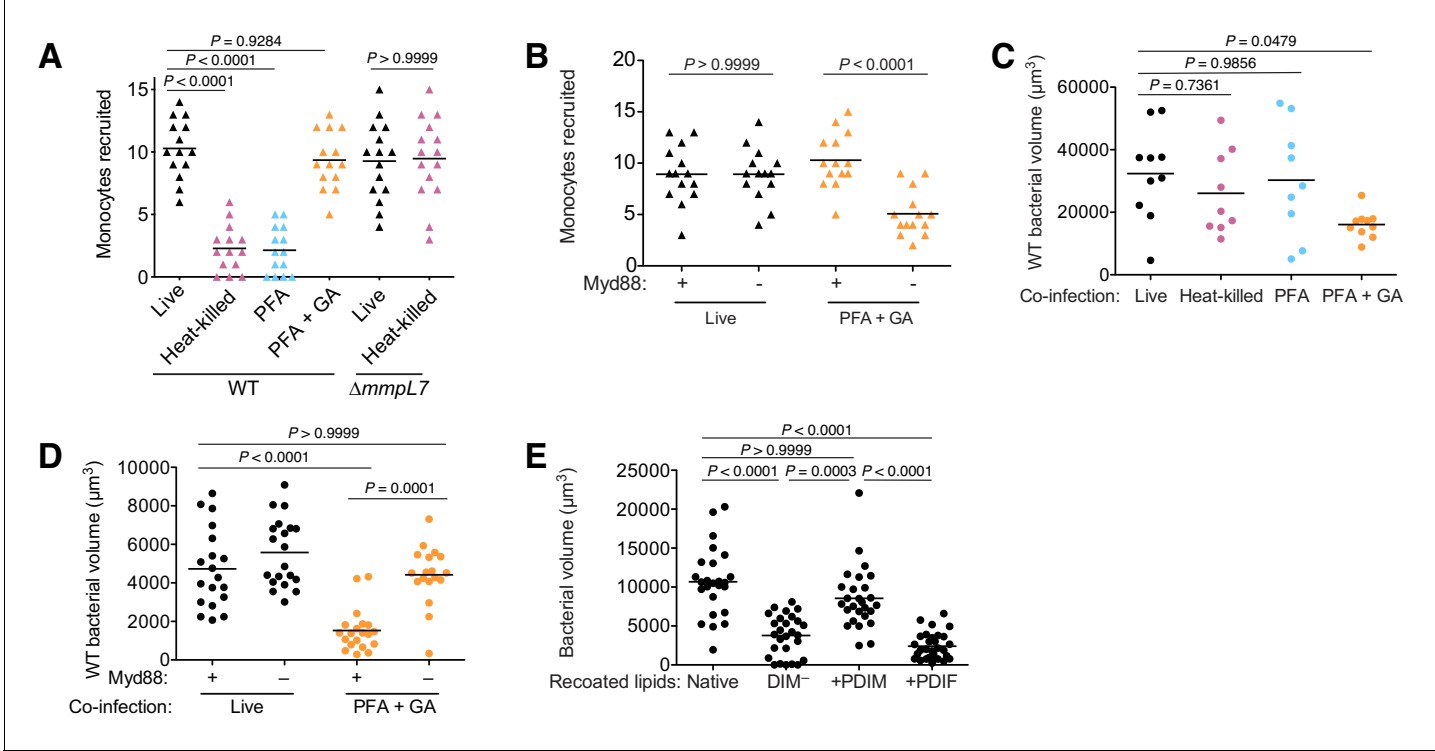

**Figure 7.** PDIM spreading into epithelial membranes is required to evade microbicidal monocytes. (A) Mean monocyte recruitment at 3 hpi of the HBV with ~100 live, heat-killed, PFA treated, or PFA+GA treated wildtype (WT) or ~100 live or heat-killed Δ*mmpL7 M. marinum*. (B) Mean monocyte recruitment at 3 hpi of the HBV of wildtype or Myd88-depleted fish with ~100 live or PFA+GA treated wildtype *M. marinum*. Mean volume of wildtype *M. marinum* following co-infection with (C) wildtype *M. marinum* treated as in A or (D) live or PFA+GA treated wildtype *M. marinum* in wildtype or Myd88-depleted fish. (E) Mean bacterial volume 3 days following HBV infection of wildtype fish with ~100 *M. marinum* recoated with Native, DIM-depleted (DIM⁻), DIM⁻ plus PDIM (+PDIM), or DIM⁻ plus PDIF (+PDIF) lipids. (A)-(C) ordinary one-way ANOVA with Tukey's multiple comparisons test with selected adjusted *P* values shown. (D) and (E) Kruskal-Wallis ANOVA for unequal variances with Dunn's multiple comparisons test with selected adjusted *P* values shown. (A)-(E) representative of three separate experiments.

The online version of this article includes the following source data for figure 7:

**Source data 1.**

recruited monocytes to a similar extent as live bacteria (*Figure 7A*). Moreover, this monocyte recruitment was now downstream of Myd88 signaling (*Figure 7B*). Taken together these data demonstrate that PDIM spreading into epithelial membranes is required to prevent Myd88-dependent recruitment of monocytes.

Not only are PDIM-deficient bacteria killed by Myd88-recruited monocytes but co-infected PDIM-expressing bacteria are as well (*Cambier et al., 2014b*). This latter finding allowed us to directly examine the link between PDIM spreading and bacterial killing by resultant Myd88-signaled monocytes. We infected larvae with live red-fluorescent wildtype *M. marinum* along with green-fluorescent wildtype *M. marinum* that were live (untreated), heat-killed, PFA treated, or PFA+GA treated and then determined the burdens of the red-fluorescent wildtype *M. marinum* at 3 dpi. Only co-infection with PFA+GA treated bacteria caused attenuation of wildtype bacteria (*Figure 7C*). Moreover, this attenuation was dependent on Myd88 (*Figure 7D*). Lastly, we asked if PDIF, which is unable to spread into epithelial cells (*Figure 6F*), could rescue DIM-deficient bacterial growth similarly to PDIM (*Figure 2F*). Consistent with the requirement for PDIM to spread into epithelial cells to promote virulence, we found that recoating with PDIF does not rescue bacterial growth (*Figure 7E*). Together these findings implicate PDIM spreading into epithelial membranes in inhibiting TLR/Myd88 signaling and the resultant recruitment of microbicidal monocytes that can kill infecting mycobacteria.

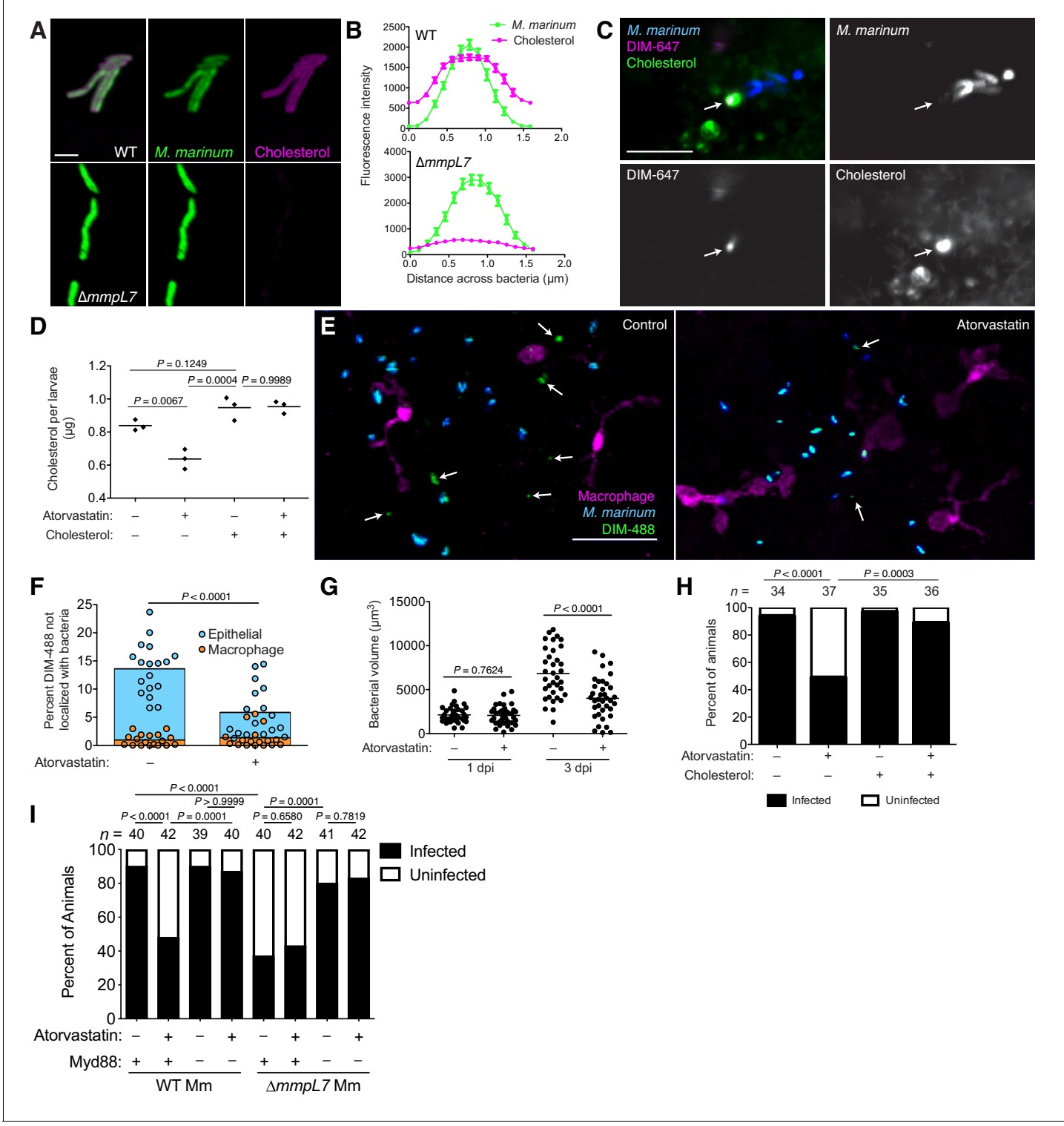

**Figure 8.** Host cholesterol promotes PDIM spread and mycobacterial infectivity. (**A**) Images of wildtype (WT) and Δ*mmpL7 M. marinum* expressing a green-fluorescent protein following 48 hr incubation with alkyne-cholesterol reacted with AlexaFlour647 Azide, scale bar = 3 μm. (**B**) Mean (± SEM) fluorescent intensity of line profiles drawn perpendicular to poles of WT and Δ*mmpL7 M. marinum* labeled as in A. (**C**) Image of A549 epithelial cells labeled with BODIPY-cholesterol at 1 dpi with DIM-647 labeled *M. marinum* at an MOI of 5. Arrows, spread DIM-647 co-localizing with BODIPY-cholesterol, scale bar = 10 μm. (**D**) Mean cholesterol content of 3 dpf zebrafish following a 24 hr treatment with atorvastatin, water-soluble cholesterol, or both. Ordinary one-way ANOVA with Tukey's multiple comparisons test with selected adjusted *P* values shown. (**E**) Images of control or atorvastatin treated transgenic fish whose macrophages express tdTomato at 2 hpi with ~100 *M. marinum* expressing a cytosolic blue-fluorescent protein recoated

*Figure 8 continued on next page*

*Figure 8 continued*

with DIBO-488 labeled azido-DIM (DIM-488), scale bar = 40 µm. Arrows, DIM-488 spread onto epithelial cells. (F) Mean percent DIM-488 in macrophage or epithelial cells not localized with bacteria at 2 h following HBV infection with ~100 *M. marinum* in control or atorvastatin treated fish. Two-tailed, unpaired t test. (G) Mean bacterial volume following HBV infection of control or atorvastatin treated fish with ~100 *M. marinum*. Two-tailed Mann Whitney test for 2 hpi and two-tailed, unpaired t test for 24 hpi. (H) Percentage of infected or uninfected fish at 3 dpi into the HBV with 1–3 wildtype *M. marinum* with or without atorvastatin and water-soluble cholesterol. (I) Percentage of infected or uninfected wildtype or Myd88-depleted fish at 3 dpi into the HBV with 1–3 wildtype or Δ*mmpL7 M. marinum* with or without atorvastatin. (H) and (I) Fisher's exact test with Bonferroni's correction for multiple comparisons. (B)-(D) and (F)-(I) representative of three separate experiments.

The online version of this article includes the following source data and figure supplement(s) for figure 8:

**Source data 1.**
**Figure supplement 1.** Cholesterol mediates DIM-488 spread on A549 epithelial cells.
**Figure supplement 1—source data 1.**

## Host cholesterol promotes PDIM spread and mycobacterial infectivity

We next asked if host lipids also influence PDIM spreading. Cholesterol has been reported to modulate macrophage interactions with mycobacteria (*Gatfield and Pieters, 2000*). However, three lines of evidence suggest a macrophage-independent relationship between mycobacteria and cholesterol: (1) 36% of the lipid extracted from *Mycobacterium bovis* harvested from necrotic mouse lung (where macrophages are sparse) was found to be host cholesterol *Kondo and Kanai, 1976*; (2) mycobacterial-associated cholesterol was isolated as a mixture with PDIM *Kondo and Kanai, 1976*; (3) *M. tuberculosis* grown in culture can sequester cholesterol to its outer mycomembrane, a process dependent on mycomembrane lipids (*Brzostek et al., 2009*). To test if host cholesterol interacted with PDIM to facilitate spreading, we first grew wildtype and Δ*mmpL7 M. marinum* in the presence of alkyne-cholesterol (*Figure 8—figure supplement 1A*) for 48 hr, followed by treatment with azide-conjugated alexafluor-647. We found that only wildtype *M. marinum* could sequester cholesterol to their mycomembrane (*Figure 8A and B*). Considering PDIM represents a substantial fraction of the mycomembrane, we wondered if the inability of Δ*mmpL7 M. marinum* to sequester cholesterol was due to a decrease in overall hydrophobicity. We grew mycobacteria in the similarly hydrophobic but structurally distinct azide-labeled phosphoethanolamine (Azido-PE, *Figure 8—figure supplement 1B*). Following reaction with DIBO-647, we found that both wildtype and Δ*mmpL7 M. marinum* could equally sequester azido-PE (*Figure 8—figure supplement 1C and D*). To determine if this PDIM-cholesterol association was also operant in mammalian membranes, we labeled A549 epithelial cells with the cholesterol surrogate BODIPY-cholesterol. Following infection with DIM-647 labeled *M. marinum* we found spread DIM-647 in areas of intense BODIPY-cholesterol signal (*Figure 8C*, arrow).

We next tested whether this cholesterol-PDIM association was promoting PDIM's ability to spread into host membranes. We depleted cholesterol from A549 epithelial cells with methyl β−cyclodextrin (MßCD) which resulted in an eight-fold decrease in cholesterol (*Figure 8—figure supplement 1E*). This was associated with decreased DIM-488 spreading (*Figure 8—figure supplement 1F*). Importantly, in MßCD treated cells we could restore cholesterol to untreated levels with water-soluble cholesterol, and this restored DIM-488 spreading (*Figure 8—figure supplement 1E and F*). To test if host cholesterol facilitates PDIM spreading in vivo, we treated zebrafish with statins, drugs that inhibit HMG-CoA reductase, the rate-limiting step of cholesterol biosynthesis. Specifically, we used atorvastatin which was shown to lower cholesterol in zebrafish (*Maerz et al., 2019*). Following 24 hr of treatment, atorvastatin decreased cholesterol levels in the larvae by 25%, and co-treatment with water-soluble cholesterol restored cholesterol levels (*Figure 8D*). Atorvastatin treatment also decreased DIM-488 spreading onto epithelial cells (*Figure 8E and F*), suggesting that PDIM's interaction with cholesterol promotes spreading into epithelial membranes in vivo.

Hypercholesteremia has been shown to exacerbate mycobacterial infection in mice and humans (*Martens et al., 2008*; *Soh et al., 2016*), and studies suggest the utility of statins both in TB treatment and prevention. When used in mice, statins decreased the duration of TB therapy by one month (*Dutta et al., 2016*), and studies of health care databases found that statin use is associated with decreased incidence of TB (*Kim et al., 2019*; *Lai et al., 2016*). Accordingly, we sought to evaluate the role of statins in the reduction of bacterial burdens and in preventing infection. First, we

showed that statin treatment reduced bacterial burdens in the zebrafish (*Figure 8G*). Next, to evaluate if statins prevented infection, we infected zebrafish with 1–3 bacteria, similar to the infectious dose in humans (*Bates et al., 1965*; *Wells et al., 1948*), and evaluated their ability to establish infection. This infectivity assay previously found that PDIM-deficient *M. marinum* established infection at a reduced frequency (*Cambier et al., 2017*). We treated zebrafish with atorvastatin for 24 hr prior to infection with 1–3 bacteria and continued daily atorvastatin treatment for the 3 day assay period. At 3 dpi, atorvastatin treatment decreased *M. marinum*'s infectivity by 50% (*Figure 8H*). Moreover, restoring cholesterol levels by co-treating with water-soluble cholesterol restored *M. marinum*'s infectivity (*Figure 8H*). This result confirmed atorvastatin reduces infectivity by lowering cholesterol rather than to off target effects. However, this protective low-cholesterol state could be due to antimicrobial effects independent of PDIM spread. Altered cholesterol flux can have pleiotropic effects on immunity (*Tall and Yvan-Charvet, 2015*), and mycobacteria have been shown to use cholesterol as a carbon source (*Pandey and Sassetti, 2008*).

To test the alternative hypothesis that atorvastatin's protection is not through disrupting PDIM spread, we evaluated the infectivity of PDIM-deficient Δ*mmpL7 M. marinum*. Consistent with previous reports (*Cambier et al., 2017*), Δ*mmpL7 M. marinum* exhibited decreased infectivity compared to wildtype bacteria (*Figure 8I*). In agreement with our model that atorvastatin acts through modulating PDIM spread, we found that atorvastatin treatment did not further decrease Δ*mmpL7*'s infectivity (*Figure 8I*). However, PDIM-deficient bacteria exhibit such a severe virulence defect, any further attenuation due to other low-cholesterol-dependent antimicrobial mechanisms may not be possible. To address this concern, we took advantage of the fact that Δ*mmpL7*'s virulence defect is reversed in the absence of host Myd88 signaling (*Figure 8I*, *Cambier et al., 2014b*). In this condition where PDIM-deficient bacteria are fully virulent, they should become attenuated by any non-PDIM-associated antimicrobial mechanisms caused by atorvastatin. However, atorvastatin still did not decrease Δ*mmpL7*'s infectivity in Myd88-depleted hosts (*Figure 8I*). Finally, we showed that atorvastatin decreased the infectivity of wildtype *M. marinum* in a Myd88-dependent fashion (*Figure 8I*). These data strongly support the idea that atorvastatin acts to disrupt the PDIM-Myd88 axis. Only in conditions where PDIM is present on bacterial surfaces and when host Myd88 signaling is intact will lowering cholesterol provide a protective effect. Collectively, these data demonstrate that statins reduce mycobacterial infectivity by reducing host cholesterol, and thereby PDIM's infiltration of epithelial membranes.

## Discussion

Molecular Koch's postulates are a guiding set of principles used to characterize bacterial virulence factors (*Falkow, 1988*; *Falkow, 2004*; *Ramakrishnan, 2020*). Genetic analyses are the mainstay of assigning pathogenic functions to specific virulence factors, including those not directly encoded by the genome. Virulence lipids are often assigned their pathogenic roles from studies of bacterial mutants lacking proteins involved in the lipid's biosynthesis or transport. Thus, these functions can be directly attributed only to the biosynthetic protein, and not the lipid per se. Here, by combining bioorthogonal chemistry with the zebrafish model of TB, we more rigorously satisfy molecular Koch's postulates to describe PDIM's role in virulence. PDIM's mobility and the abundance of cholesterol in epithelial membranes influenced PDIM's ability to infiltrate the host lipid environment. This occupation of epithelial membranes by PDIM mediated subversion of TLR/Myd88 signaling at the site of infection, thus enabling mycobacteria to gain access to non-activated immune cells (*Figure 9*).

All of our insights into PDIM biology were made using our chemical extraction and recoating approach. While we do not know if the mycomembrane organization of the recoated bacteria mimics that of untouched bacteria, our data do demonstrate that these bacteria are indistinguishable with regards to their virulence. Additionally, our work is in agreement with results evaluating native PDIM from *M. tuberculosis*, where PDIM was found to occupy macrophage membranes (*Augenstreich et al., 2019*) and disrupt membrane protein signaling (*Augenstreich et al., 2020*). Thus, we believe the data generated by our approach represents native PDIM biology. In addition to being able to monitor PDIM's distribution during in vivo infection, our approach allowed us to observe and correlate lipid mobility and membrane fluidity on the bacterial surface to the ability of lipids to spread into host membranes. While both TMM and PDIM spread into macrophage membranes, only PDIM was able to occupy epithelial membranes following infection. PDIM's increased

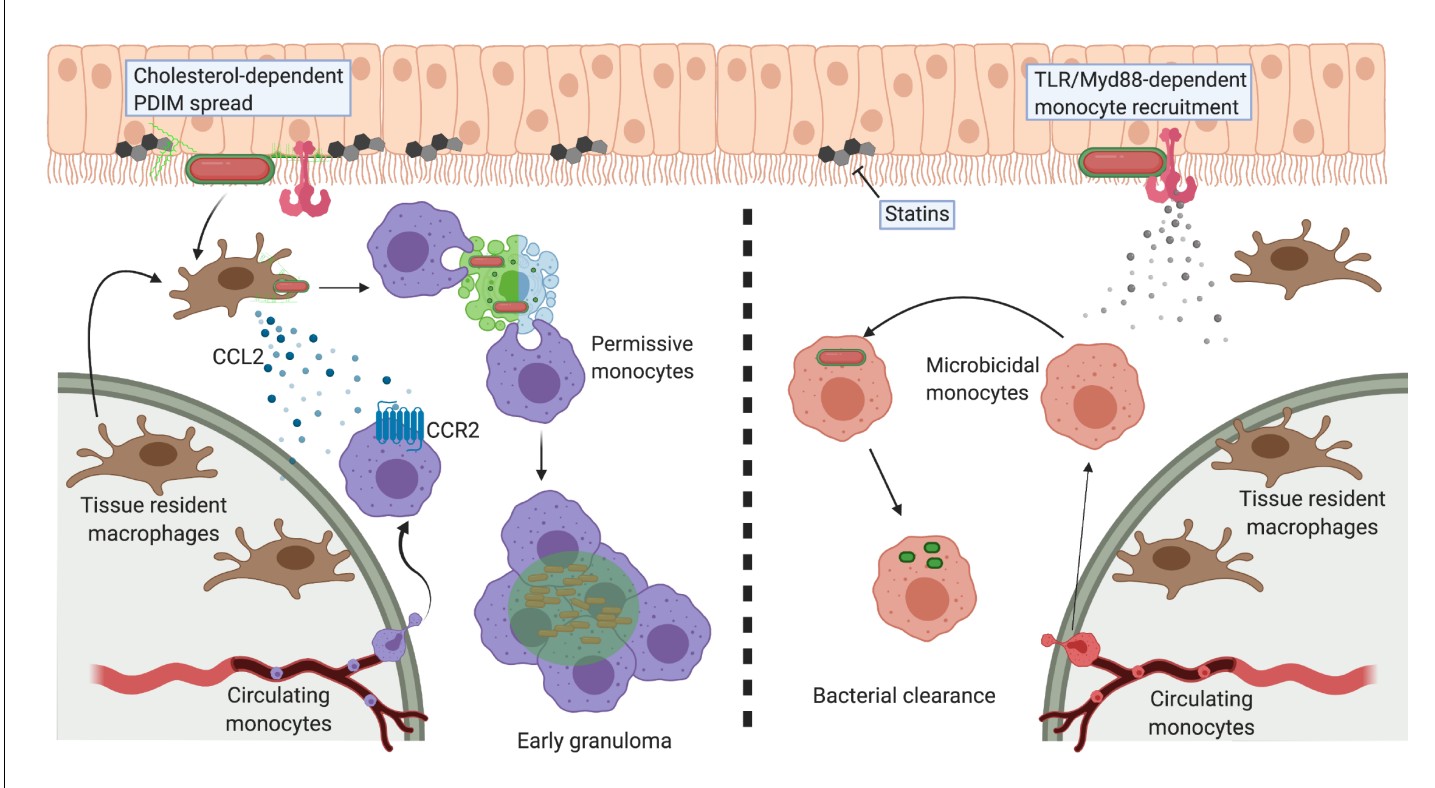

**Figure 9.** Model of PDIM spreading to promote virulence in the zebrafish hindbrain ventricle. Cholesterol-dependent PDIM spread into epithelial membranes prevents TLR/Myd88 detection at the site of infection. Bacteria then drive resident macrophages to produce CCL2 to recruit CCR2-positive permissive monocytes which go on to form early granulomas. PDIM continues to spread into host membranes throughout this process. In low cholesterol settings (statin treatment) PDIM does not spread as readily into epithelial membranes. TLR/Myd88-dependent recruitment of microbicidal monocytes occurs, which can then clear infecting mycobacteria. Figure created with BioRender.com.

mobility in comparison to TMM could be promoting an active spread through host membranes or, alternatively, could be promoting membrane mixing between bacteria and host.

Previous models suggest PDIM facilitates a passive evasion of TLRs by masking underlying TLR ligands (*Cambier et al., 2014b*; *Cambier et al., 2017*). This model was based on findings that PDIM-expressing mycobacteria were rendered growth-attenuated if co-infected with TLR-stimulating bacteria (*Cambier et al., 2014b*). While our current model suggests a more active role for PDIM, we still find that TLR-stimulating bacteria cause attenuation of wildtype bacteria. Specifically, bacteria unable to spread PDIM caused attenuation of bacteria that can spread PDIM (*Figure 7C and D*). These findings argue that PDIM's influence over innate immune signaling is spatially restricted. PDIM appears to only inhibit TLR/Myd88 signaling at membrane sites where it has spread, emphasizing the need for PDIM localization on the bacterial surface prior to spreading into epithelial membranes.

Cholesterol plays a multifaceted role in mycobacterial pathogenesis. Here we describe its role in promoting mycobacterial infectivity through promoting PDIM spread into epithelial membranes. This mechanism of pathogenesis is particularly interesting as mycobacteria can utilize cholesterol as a carbon source to bolster PDIM production (*Wilburn et al., 2018*). Systemic cholesterol metabolism in humans influences the abundance of cholesterol in lung epithelial membranes (*Fessler, 2017*). Therefore, cholesterol-promoted PDIM spreading may also influence *M. tuberculosis* transmission. While a clinical trial is currently underway evaluating the use of statins alongside standard TB therapy (*Karakousis et al., 2019*), our data instead argue for the use of statins as a TB preventative therapy.

# Materials and methods

## Resources table

**Key resources table**

| Reagent type (species) or resource | Designation | Source or reference | Identifiers | Additional information |
|---|---|---|---|---|
| Strain, strain background (*Mycobacterium marinum* M strain) | wildtype *M. marinum* | *Takaki et al., 2013*. | derivatives of ATCC #BAA-535 | Fluorescent strains: pMS12:tdTomato, pMS12:EBFP2, or pMS12:wasabi |
| Strain, strain background (*M. marinum* M strain) | Δ*mmpL7 M. marinum* | *Cambier et al., 2014b*. | NA | Fluorescent strains: pMS12:tdTomato, pMS12:wasabi |
| Strain, strain background (*Escherichia coli*) | Top10 | *Urbanek et al., 2014*. | NA | Transformed with pBAD:tret from *Thermoproteus tenax* |
| Genetic reagent (Zebrafish, *Danio rerio*) | Wildtype AB | Stanford University | ZFIN ID: ZDB-GENO -960809–7 | |
| Genetic reagent (Zebrafish, *Danio rerio*) | *Tg(mfap4:tdTomato)* | *Walton et al., 2015*. | ZFIN ID: ZDB-TGCONSTRCT -160122–3 | |
| Genetic reagent (Zebrafish, *Danio rerio*) | *Tg(flk1:mcherry)* | *Wang et al., 2010*. | ZFIN ID: ZDB-TGCONSTRCT -110127–23 | |
| Chemical compound, drug | Iodotrimethylsilane | ACROS | Cat#10530964 CAS:16029-98-4 | |
| Chemical compound, drug | Sodium azide | Sigma-Aldrich | Cat#S8032 CAS: 26628-22-8 | |
| Chemical compound, drug | Tetrabutylammonium hydroxide solution | Sigma-Aldrich | Cat#178780 CAS: 2052-49-5 | |
| Chemical compound, drug | Lignoceric acid | Sigma-Aldrich | Cat# L6641 CAS: 557-59-5 | |
| Chemical compound, drug | Click-IT Alexa Fluor 488 DIBO Alkyne | Thermo-Fisher | Cat# C10405 Discontinued* | |
| Chemical compound, drug | Click-IT Alexa Fluor 647 DIBO Alkyne | Thermo-Fisher | Cat# C10408 Discontinued* | |
| Chemical compound, drug | Alexa Fluor 647 Azide | Thermo-Fisher | Cat# A10277 | |
| Chemical compound, drug | Alkyne-Cholesterol | Click Chemistry Tools | Cat# 1409 | |
| Chemical compound, drug | 6-azido-6-deoxy-glucose | Sigma-Aldrich | Cat# 712760 CAS: 20847-05-6 | |
| Chemical compound, drug | UDP-glucose | Sigma-Aldrich | Cat# U4625 CAS: 28053-08-9 | |
| Chemical compound, drug | Alexa-647 hydroxylamine | Thermo-Fisher | Cat# A30632 | Chemical |
| Compound, drug | Methyl-ß-cyclodextrin | Sigma-Aldrich | Cat# C4555 CAS: 128446-36-6 | |
| Chemical compound, drug | Cholesterol-water soluble | Sigma-Aldrich | Cat# C4951 | |
| Chemical compound, drug | Atorvastatin | Sigma-Aldrich | Cat# PHR1422 | |
| Chemical compound, drug | Methylene blue | Sigma-Aldrich | Cat# M9140 CAS: 122965-43-9 | |
| Chemical compound, drug | Tango Buffer (10x) | Thermo-Fisher | Cat#BY5 | |
| Chemical compound, drug | Phenol Red Sodium Salt | Sigma-Aldrich | Cat#P4758 CAS: 34487-61-1 | |

*Continued on next page*

*Continued*

| Reagent type (species) or resource | Designation | Source or reference | Identifiers | Additional information |
|---|---|---|---|---|
| Chemical compound, drug | PMA (Phorbol 12-myristate 13-acetate) | Sigma-Aldrich | Cat#P1585 CAS:16561-29-8 | |
| Chemical compound, drug | Hygromycin B | Thermo-Fisher | Cat#10687010 | |
| Chemical compound, drug | Glutaraldehyde solution (70% in water) | Sigma-Aldrich | Cat#G7776 CAS: 111-30-8 | |
| Chemical compound, drug | 16% paraformaldehyde solution | Thermo-Fisher | Cat#28908 | Chemical |
| Compound, drug | BTTP | Click Chemistry Tools | Cat#1414 CAS: 1334179-85-9 | |
| Chemical compound, drug | 16:0 azidocaproyl phosphoethanolamine (Azido-PE) | Avanti Polar Lipids | Cat#870126 CAS: 2252461-34-8 | |
| Chemical compound, drug | BODIPY-Cholesterol | Cayman Chemical | Cat#24618 CAS: 878557-19-8 | |
| Chemical compound, drug | Alexa-Fluor 594 Wheat Germ Agglutinin | Fisher | Cat#W11262 | |
| Cell line (*Homo sapiens*) | A549 human alveolar epithelial cells | ATCC | CCL-185 | |
| Cell line (*Homo sapiens*) | THP-1 human monocytic cell line | ATCC | TIB-202 | |
| Sequence-based reagent | *myD88* morpholino sequence: GTTAAACACTGAC CCTGTGGATCAT | *Bates et al., 2007*. | ZFIN ID: ZDB-MRPHLNO -080325–4 | |
| Commercial assay or kit | Total Cholesterol and Cholesterol Ester Colorimetric/Fluorometric Assay Kit | Bio-Vision | Cat# K603 | |
| Software, algorithm | NIS-Elements | Nikon | | |
| Software, algorithm | Imaris | Bitplane | | |
| Software, algorithm | Prism | GraphPad | | |

   *Since preparing this manuscript we have found that AFDye 488 DBCO (Click Chemistry Tools, Cat# 1278) and AFDye 647 DBCO (Click Chemistry Tools, Cat# 1302) are suitable alternatives to the DIBO fluorophores. The DBCO dyes can be used with the same concentrations and labeling protocols presented in this manuscript for the DIBO dyes (Unpublished data).

## Procedures and materials for synthetic chemistry

All reactions were performed in dry standard glassware fitted with rubber septa under an inert atmosphere of nitrogen unless otherwise stated. Preparative thin-layer chromatography (TLC) was performed with Millipore's 1 mm and 0.2 mm silica gel 60 pre-coated glass plates. Analytical TLC was used for reaction monitoring and product detection using pre-coated glass plates covered with 0.20 mm silica gel with fluorescent indicator; visualized by UV light and 10% CuSO4 in 1.3M phosphoric acid in water. Reagents were purchased in reagent grade from commercial suppliers and used as received, unless otherwise described. Anhydrous dichloromethane (DCM) was prepared by passing the solvent through an activated alumina column.

## Chemical analysis instrumentation

Proton ($^1$H NMR) and proton-decoupled carbon-13 ($^{13}$C {$^1$H} NMR) nuclear magnetic resonance spectra (*Supplementary file 2*) were recorded on an Inova-500 spectrometer at 25℃, are reported in parts per million downfield from tetramethylsilane, and are referenced to the residual protium (CDCl$_3$: 7.26 [CHCl$_3$]) and carbon (CDCl$_3$: 77.16) resonances of the NMR solvent. Data are

represented as follows: chemical shift, multiplicity (br = broad, s = singlet, d = doublet, t = triplet, q = quartet, quin = quintet, sept = septet, m = multiplet), coupling constants in Hertz (Hz), integration. Mass spectra were obtained on a Bruker Microflex MALDI-TOF by mixing 0.5 µl of 1 mg/ml sample in chloroform with 0.5 µl of 10 mg/ml 2,5-dihydroxybenzoic acid before spotting onto a 96-well MALDI plate.

## Isolation of PDIM

**Chemical structure 1.** PDIM.

Wildtype *M. marinum* was grown in GAS medium without Tween-80 to an $OD_{600}$ of 1.2. Bacteria were pelleted, frozen and lyophilized on a Labconoco FreezeZone 4.5 plus. Bacterial lipids were then extracted by stirring lyophilized pellet in petroleum ether for 1 hr at 23°C. Bacteria were allowed to settle, and solvent was collected. Remaining bacteria were re-extracted 3–5 times. The solvent extract was then passed through a 0.2 µm PTFE filter, and crude lipids were concentrated under reduced pressure. Crude lipid extracts were separated by preparative TLC in the solvent system 98:2 petroleum ether:ethyl acetate. The band correlating to PDIM at an Rf = 0.4 was isolated. Preparative TLC was repeated twice more to further purify PDIM. Average PDIM isolated from *M. marinum* was 60 mg per 10 liters of culture. [1]H NMR (500 MHz, CDCl3) δ 4.93–4.87 (m, 2H), 3.32 (s, 3H), 2.87–2.83 (m, 1H), 2.57–2.51 (m, 2H), 1.90–0.81 (m, 166H). [13]C NMR (126 MHz, CDCl3) δ 176.69, 176.63, 86.78, 86.73, 77.41, 77.37, 77.16, 76.90, 70.72, 70.67, 57.51, 45.56, 45.51, 41.41, 41.33, 38.56, 37.85, 37.73, 37.09, 36.77, 34.91, 34.18, 34.07, 32.83, 32.08, 30.24, 29.96, 29.87, 29.82, 29.75, 29.62, 29.53, 28.34, 27.64, 27.36, 27.25, 27.16, 27.12, 25.74, 25.35, 22.85, 22.74, 22.43, 20.81, 20.57, 20.52, 20.37, 20.27, 18.67, 18.58, 18.53, 14.80, 14.75, 14.28, 14.22, 10.26, 1.16. MALDI-TOF for: C82H162O5 Calc'd [M+Na+]=1250.23; found 1250.41. C84H166O5 Calc'd [M+Na+]=1278.26, found 1278.40. C86H170O5 Calc'd [M+Na+]=1306.29, found 1306.55.

## Synthesis of azido-DIM

**Chemical structure 2.** Azido-DIM synthesis.

To PDIM (75.6 mg, 60.2µmol, 1.0 equiv.) was added 0.3 ml of DCM, the mixture was stirred and iodotrimethylsilane (TMSI, 258 µl, 1.8 mmol, 30.0 equiv.) was added and allowed to react for 12 hr at 23°C. The reaction was concentrated under reduced pressure. NaN3 (39.1 mg, 602µmol, 10.0 equiv.) was added followed by 0.3 ml of anhydrous dimethylformamide (DMF) and stirred for 12 hr at 23°C. The solvent was removed under reduced pressure and the product was purified using

preparative TLC in 98:2 petroleum ether:ethyl acetate Rf = 0.5 as a white wax (37.7 mg, 29.8µmol, 50%). $^1$H NMR (500 MHz, CDCl$_3$) δ 5.06–4.86 (m, 2H), 3.30–3.21 (m, 1H), 2.58–2.50 (m, 2H) 2.1–0.81 (m, 163H). $^{13}$C NMR (126 MHz, CDCl$_3$) δ 176.73, 176.63, 77.41, 77.36, 77.16, 76.91, 71.08, 70.74, 59.86, 45.56, 41.96, 41.45, 41.35, 39.27, 37.84, 37.73, 37.24, 37.08, 37.01, 35.28, 34.82, 34.18, 32.08, 30.32, 30.24, 30.22, 30.04, 29.96, 29.87, 29.82, 29.75, 29.72, 29.67, 29.62, 29.52, 29.39, 28.40, 28.34, 27.72, 27.36, 27.25, 27.18, 27.16, 26.20, 25.35, 25.27, 25.23, 22.85, 21.28, 21.13, 20.35, 20.32, 20.27, 18.73, 18.66, 18.58, 18.53, 14.56, 14.29, 13.87. MALDI-TOF for: C81H159N3O4 Calc'd [M+Na$^+$]=1261.22; found 1260.92. C83H163N3O4 Calc'd [M+Na$^+$]=1289.25, found 1289.02. C85H167N3O4 Calc'd [M+NH4$^+$]=1312.33, found 1312.37.

## Hydrolysis of PDIM and isolation of phthiocerol

**Chemical structure 3.** PDIM hydrolysis.

To dried PDIM (66.7 mg, 53.1µmol) was added 0.5 ml of 40% tetrabutylammonium hydroxide in water, the mixture was sealed, aggressively stirred, and allowed to react for 48 hr at 120°C. Reaction was cooled to 23°C and 4M HCL was added dropwise until pH was <3. The reaction was extracted with DCM and washed once with water. The solvent was removed under reduced pressure and phthiocerol was purified using preparative TLC in 80:20 petroleum ether:ethyl acetate Rf = 0.4. (18.8mg, 42.5µmol, 80%). $^1$H NMR (500 MHz, CDCl$_3$) and $^{13}$C NMR (126 MHz, CDCl$_3$) matched previously reported spectra. ESI HRMS for: C28H58O3 Calc'd [M+H$^+$]=443.4464; found 443.4449. C29H60O3 Calc'd [M+H$^+$]=457.4621; found 457.4616. C30H62O3 Calc'd [M+H$^+$]=471.4777, found 471.4762.

## Synthesis of PDIF

**Chemical structure 4.** PDIF synthesis.

To phthiocerol (18.8 mg, 42.5 µmol, one equiv.) was added lignoceric acid (47 mg, 127.5 µmol, 3.0 equiv.), N,N'-dicyclohexylcarbodiimide (DCC, 35 mg, 170µmol, 4.0 equiv.), 4-Dimethylaminopyridine (DMAP, 20.77 mg, 170 µmol, 4.0 equiv.), and 2.3 ml of DCM. The reaction was stirred and allowed to react for 16 hr at 23°C. The solvent was removed under reduced pressure and the product was purified using preparative TLC in 98:2 petroleum ether:ethyl acetate Rf = 0.4. (12.8 mg, 11.5 µmol, 27%). $^1$H NMR (500 MHz, CDCl$_3$) δ 4.94–4.87 (quin, 2H), 3.33 (s, 3H), 2.88–2.83 (m, 1H), 2.30–2.25 (t, 4H), 1.75–0.79 (m, 141H). $^{13}$C NMR (126 MHz, CDCl$_3$) δ 173.42, 86.66, 77.27, 77.02, 76.76, 70.97, 57.40, 38.45, 34.84, 34.67, 34.13, 34.09, 32.62, 31.94, 29.73, 29.68, 29.55, 29.47, 29.38, 29.34, 29.23, 27.44, 25.57, 25.17, 25.10, 22.70, 22.37, 14.76, 14.12, 10.10. MALDI-TOF for: C76H150O5 Calc'd [M+Na$^+$]=1166.14, found 1166.14. C77H152O5 Calc'd [M+Na$^+$]=1180.15, found 1180.24. C78H154O5 Calc'd [M+Na$^+$]=1194.17; found 1194.25.

## Synthesis of azido-DIF

**Chemical structure 5.** Azido-DIF synthesis.

To PDIF (12.8 mg, 11.5 µmol, one equiv.) was added 0.3 ml of DCM, the mixture was stirred and TMSI (49 µl, 345µmol, 30.0 equiv.) was added and allowed to react for 3 hr at 23°C. The reaction was concentrated under reduced pressure. NaN$_3$ (7.5 mg, 115µmol, 10.0 equiv.) was added followed by 0.3 ml of anhydrous dimethylformamide (DMF) and stirred for 12 hr at 23°C. The solvent was removed under reduced pressure and the product was purified using preparative TLC in 98:2 petroleum ether:ethyl acetate Rf = 0.5. (6.9mg, 6µmol, 52%). $^1$H NMR (500 MHz, CDCl$_3$) δ 5.08–5.00 (m, 2H), 3.32–3.20 (m, 2H), 3.12–3.04 (m, 1H), 2.33–2.25 (t, 4H), 1.80–0.73 (m, 138H). $^{13}$C NMR (126 MHz, CDCl$_3$) δ 173.41, 77.27, 77.02, 76.76, 71.16, 70.05, 59.72, 39.09, 36.92, 35.12, 34.70, 34.64, 32.34, 31.94, 29.72, 29.67, 29.58, 29.53, 29.42, 29.38, 29.33, 29.23, 26.05, 25.15, 24.02, 22.70, 21.15, 16.10, 14.13, 11.06, 1.03. ESI C75H148NO4 calc'd [M+K, -N$_2$]=1166.10, found 1166.05.

## Zebrafish husbandry and infections

Wildtype AB (Zebrafish International Resource Center), and *Tg(mfap4:tdTomato)* (*Walton et al., 2015*) lines were maintained in buffered reverse osmotic water systems. Fish were fed twice daily a combination of dry feed and brine shrimp and were exposed to a 14 hr light, 10 hr dark cycle to maintain proper circadian conditions. Zebrafish embryos were maintained at 28.5°C in embryo media which consisted of the following dissolved in Milli-Q water (%weight/volume): 0. 0875% sodium chloride, 0.00375% potassium chloride, 0.011% calcium chloride, 0.00205% monopotassium phosphate, 0.00089% disodium phosphate, and 0.0493% magnesium sulfate. Embryo media was then buffered to pH 7.2 with sodium bicarbonate. Embryos were maintained in 0.25 mg/ml methylene blue (Sigma) from collection to 1 day post-fertilization (dpf). 0.003% PTU (1-phenyl-2-thiourea, Sigma) was added from 24 hr post-fertilization (hpf) on to prevent pigmentation. Larvae (of undetermined sex given the early developmental stages used) were infected at 48 hpf via the hindbrain ventricle (HBV) using single-cell mycobacterial suspensions of known titer. Number of animals to be used for each experiment was guided by pilot experiments or by past results with other bacterial mutants and/or zebrafish. On average 15 to 40 larvae per experimental condition were required to reach statistical significance and each experiment was repeated at least three times. Larvae were randomly allotted to the different experimental conditions. The zebrafish husbandry briefly described above and all experiments performed on them were in compliance with the U.S. National Institutes of Health guidelines and approved by the Stanford Institutional Animal Care and Use Committee.

## Bacterial strains and methods

*M. marinum* strain M (ATCC BAA-535) and Δ*mmpL7* mutants (*Cambier et al., 2014b*) expressing either TdTomato, Wasabi, or EBFP2 under the control of the *msp12* promoter were grown under hygromycin (Thermo-Fisher) selection in 7H9 Middlebrook's medium (Difco) supplemented with 10% OADC (Fisher), 0.2% glycerol, and 0.05% Tween-80 (Sigma). Where noted bacteria were also grown in glycerol-alanine-salts (GAS) medium, recipe (%weight/volume) in 18 mM sodium hydroxide in Milli-Q water pH 6.6 +/- 0.05% Tween-80: 0.03% BactoCasitone (BD Science), 0.005% ferric ammonium citrate (Sigma), 0.4% potassium phosphate dibasic anhydrous (VWR), 0.2% citric acid, anhydrous (VWR), 0.1% L-alanine (Sigma), 0.12% magnesium chloride, heptahydrate (VWR), 0.06% potassium sulfate (VWR), 0.2% ammonium chloride (VWR), and 1% glycerol. To prepare heat-killed *M. marinum*, bacteria were incubated at 80°C for 20 min. To prepare fixed bacteria, bacteria were incubated in described concentrations of glutaraldehyde (Sigma) and/or paraformaldehyde (Thermo-Fisher) for 1 hr at 23°C, followed by three washes with PBS prior to experimental use.

## Cell lines and infections

All cell lines were from ATCC and were used at passages < 10. Cell lines were authenticated by the supplier and were verified to be mycoplasma negative by Lonza MycoAlert Mycoplasma Detection Assay. Cells were grown in T75 flasks (Thermo-Fisher) and maintained at 37°C and 5% CO2. THP-1 were grown in RPMI supplemented with 10% fetal bovine serum (FBS). A549 were grown in DMEM supplemented with 10% FBS. Two days prior to infection, THP-1s were plated on 8-well Nunc Lab-Tek II chamber slides (Thermo-Fisher) at a density of 150,000 cells per well with 100 nM PMA (phorbol 12-myristate 13-acetate, Sigma) in growth media and incubated at 37°C and 5% CO2 for 24 hr. One day prior to infection, the media on THP-1s was replaced with fresh growth media and the cells were incubated at 33°C and 5% CO2 for 24 hr, cells were then infected with *M. marinum* at a multiplicity of infection (MOI) of 2 in growth media. Infection was allowed to progress for 6 hr prior to washing twice with PBS and replacing with growth media. THP-1s were then incubated for 24 hr at 33°C and 5% CO2 prior to experimental end point. One day prior to infection, A549s were plated on 8-well chamber Nunc Lab-Tek II chamber slides (Thermo-Fisher) at a density of 50,000 cells per well in growth media and incubated at 37°C and 5% CO$_2$. For cholesterol labeling, BODIPY-cholesterol (Cayman Chemical) was added at 3 µg/ml one day prior to infection and cells were incubated overnight at 37°C and 5% CO2. Cells were then washed three times with PBS prior to infection. Day of infection cells were moved to 33°C and 5% CO$_2$ for 3 hr, and then were infected with *M. marinum* at a MOI of 5 and incubated at 33°C and 5% CO$_2$ for 24 hr. Infected THP-1s and A549s were then imaged as described below.

## Extraction and recoating of *M. marinum*

### Extraction

1 liter of *M. marinum* were grown in GAS medium plus Tween-80 to an OD$_{600}$ of 1.2. Bacteria were pelleted in a glass 50 ml conical tubes (Fisher) of known weight, frozen and lyophilized. The dry bacteria in 50 ml conical tubes were then weighed and the dry bacterial weight was calculated. 25 ml of petroleum ether were then added to the bacteria, and the conical tube was capped with a PTFE lined lid (Sigma) and the suspension was vortexed for 3 min. An additional 25 ml of petroleum ether was added and the sample was centrifuged for 3 min at 1000xg at 4°C. The extract was then saved or discarded depending on the downstream experimental applications and the bacteria were extracted once more as above. For total lipid extractions (not used for recoating), bacteria were treated with 1:1 chloroform:methanol for 12 hr at 60°C. Extracts were filtered, back extracted with water to remove water-soluble contaminants, dried under reduced pressure and used for downstream experiments.

### Recoating

Prior to extractions the following calculation was used to determine the amount of lipids to add back to each pellet.

(23 mg lipid/gram dry bacteria)*(weight in grams of dry bacteria)/(0.75 recoating efficiency).

Where 23 mg of lipid per gram of dry bacteria is the experimentally determined average amount of lipid removed during petroleum ether extraction of bacteria grown in these conditions. (*Figure 1—figure supplement 1B*, initial extraction) and the 0.75 recoating efficiency was also experimentally determined (*Figure 1—figure supplement 1C*). If recoating with DIM variants, 30% of the above weight consists of DIMs and the remainder will consist of DIM-depleted petroleum ether lipid extracts. Following two rounds of petroleum ether extraction as detailed above, bacteria were immediately mixed with pre-determined lipid mixtures (or no lipid at all for delipidated bacteria) in petroleum ether (2–3 ml) followed by extended drying under reduced pressure. Dried bacteria were then rescued into 7H9 media supplemented with 10% OADC, and 0.2% glycerol (prep media) and subjected to single cell preparation protocol.

## *M. marinum* single cell preparation

For more thorough details, as well as rationale and explanation of the following protocol see *Takaki et al., 2013*. Bacteria were washed once with 15 ml of prep media followed by resuspension in 500 µl of prep media. Bacteria were then passed through a 27-gauge needle 10 times, followed by the addition of 1 ml of prep media and centrifugation at 100xg for 3 min. 1 ml of supernatants

were saved. This process was repeated 3–5 times. Collected supernatants were then passed through a 5.0 µm acrodisc versapor membrane syringe filter (VWR). The filtrate was then pelleted at 16,000xg for 2 min, pellets were resuspended in prep media to a concentration of around $1 \times 10^{8-9}$ bacteria per ml, aliquoted and stored at −20C for future use, or immediately subjected to copper-free click chemistry reactions.

## Metabolic labeling of *M. marinum* with 6-TreAz

### Expression and purification of TreT

TreT was expressed and purified utilizing a similar method as previously reported (Urbanek et al., 2014). Top10 *E. coli* expressing the *tret* gene from *Thermoproteus tenax* (pBAD plasmid, AraC control) were streaked onto a Lysogeny broth (LB) agar plate supplemented with 100 µg/ml ampicillin and incubated at 37°C for 24 hr. A single colony was picked and used to inoculate 5 ml of LB liquid medium containing 100 µg/ml ampicillin. The starter culture was placed in a shaking incubator (175 rpm) at 37°C for 16 hr. The starter culture was then transferred to a 1 L solution of sterilized Terrific broth (TB) supplemented with 100 µg/ml ampicillin in a sterilized 2 L Fernbach culture flask. The flask was shaken (175 rpm) at 37°C. When the absorbance at 600 nm reached between 0.6 and 0.9 (typically 4–5 hr post inoculation), TreT expression was induced by adding 1 ml of 1 M arabinose solution in sterile water (1 mM final concentration). The flask was again shaken (175 rpm) at 37°C for another 20 hr. The culture was then transferred to a polypropylene bottle and pelleted for 15 min at 4000 x g at 4°C. The supernatant was discarded, and the pellet was suspended in 45 ml of lysis buffer (50 mM $NaH_2PO_4$, 500 mM NaCl, 20 mM imidazole, pH 7.4) and run through a homogenizer five times utilizing ice to keep the solution cool. Once homogenized, the lysate was clarified by centrifugation pelleting for 20 min at 21,000 x g at 4°C and filtering through a 0.45 µm Teflon syringe filter. To the clarified lysate (generally about 50 mg/ml protein content as measured with absorbance at 280 nm) was then added 5 ml of pre-washed (lysis buffer) Ni-NTA resin slurry (Qiagen). The suspension was mixed on an orbital shaker for 60–90 min at 4°C and then transferred to a glass column (BIO-RAD EconoColumn). Non-His-Tagged proteins were eluted with lysis buffer until the absorbance at 280 nm matched background levels utilizing 50–100 ml of lysis buffer. His-tagged TreT was eluted with elution buffer (50 mM $NaH_2PO_4$, 500 mM NaCl, 250 mM imidazole, pH 7.4) in multiple 2.5 ml increments, until protein elution was determined complete by absorbance at 280 nm. Buffer exchange to a storage/reaction buffer (50 mM Tris, 300 mM NaCl, pH 8.0) was performed with a desalting column (PD-10, GE Healthcare) and the protein was transferred to a conical tube and diluted to 1 mg/ml as determined by absorbance at 280 nm for long-term storage. Storage of the enzyme at 4°C and at this concentration yields active protein that does not have significant losses in activity even after 12 months of storage under these conditions.

### Synthesis and purification of 6-Azido-6-deoxy-trehalose (6-TreAz)

To a 1.5 ml Eppendorf tube was added 200 µl of a 100 mM solution of 6-azido-6-deoxy-glucose (final concertation 20 mM), 100 µl of a freshly-prepared 400 mM solution of UDP-glucose (final concentration 40 mM), 100 µl of a 200 mM solution of $MgCl_2$ (final concentration 20 mM), and 300 ml of reaction buffer (50 mM Tris, 300 mM NaCl, pH 7.4). Then, 300 µl of TreT solution in storage/reaction buffer was added (final concentration 300 µg/ml) and the reaction vessel was closed and mixed gently by inverting the tube. The reaction was heated to 70°C for 60–90 min with shaking at 400 rpm and then cooled on ice before further manipulation. The reaction contents were transferred to a pre-washed Amicon centrifugal filter with a nominal molecular weight limit (NWML) of 10 kDa. The filter was washed with DI water (2 × 1 ml) to facilitate maximal recovery. The upper chamber was discarded and 1 g of prewashed mixed-bed ion-exchange resin (DOWEX) was added to the filtrate and the slurry was equilibrated for 60–90 min. The suspension was filtered, and the resin was rinsed with 3–5 ml of DI water. Analysis by TLC (5:3:2 *n*-butanol: ethanol: DI water) and/or LC-MS equipped with a Supelco aminopropyl column [4.6 × 250 mm, 5 µm] (isocratic 80% ACN in DI water, 0.5 mll/min flowrate) indicated full and complete conversion of 6-azido-6-deoxy-glucose to 6-azido-6-deoxy-trehalose with high purity as determined by nuclear magnetic resonance (NMR), which matched previously reported spectra.

Synthesis and direct utilization of 6-TreAz for labeling *M. marinum*

To facilitate easier production and labeling of *M. marinum* with 6-TreAz, the reaction was performed as above but 6-TreAz was not isolated prior to labeling. Upon reaction completion, the reaction was cooled on ice to bring the vessel back to ambient temperature. The 10 mM stock of 6-TreAz was then added to cultures of *M. marinum* to a final concentration of 50 µM.

## Periodate-hydroxylamine staining of mycobacterial surfaces

Surface-exposed terminal oxidizable carbohydrates were labeled with hydroxylamine following periodate oxidation (*Beatty et al., 2000*). Lyophilized control or recoated *M. marinum* were washed twice with PBS and resuspended in 0.1 M sodium acetate (Sigma), pH 5.5 containing 1 mM sodium periodate (Sigma). Following a 20 min incubation at 4℃ with gentle rotation, 0.1 mM glycerol was added to stop the reaction. Cultures were washed three times with PBS and then subjected to single cell preparation. Following single cell preparation, pellets were transferred to a 96-well plate and then incubated with PBS containing 1 mM Alexa-647 hydroxylamine (Thermo-Fisher). Following a 2 hr incubation at 23℃, the cultures were washed five times with PBS and twice with prep media.

## Copper-free click chemistry of *M. marinum*

Following recoating or metabolic labeling, bacteria were treated for single cell preparation. Bacteria were then transferred to 96-well v-bottom dishes and were washed twice with PBS using centrifugation at 3000xg for 3 min between washing. Bacteria were then stained with either 5 µM DIBO-488 (Thermo-Fisher), or 30 µM DIBO-647 (Thermo-Fisher) in 200 µl PBS for 90 min at 23℃ protected from light. Bacteria were then washed five times in PBS followed by two washes in prep media. Bacteria were then aliquoted and stored at −20℃ for future use. Staining efficiency was evaluated by flow cytometry on a BD-Accuri C6 Plus and analysis was performed using the FlowJo software package. Staining was also evaluated by microscopy with a 60x oil-immersion Plan Apo 1.4 NA objective on the Nikon A1R confocal microscope.

## Myd88 morpholino and liposome injections

To generate Myd88 knockdown zebrafish larvae, the Myd88 morpholino 5′GTTAAACACTGACCCTG TGGATCAT3′ (*Bates et al., 2007*) was diluted to 2 mM in 0.5 x tango buffer (Thermo Scientific), containing 2% phenol red sodium salt solution (Sigma). 1 nl of the morpholino mixture was injected into the 1–4 cell stage of the developing embryo. Lipo-PBS and lipo-clodronate (http://clodronateliposomes.org) were diluted 1:10 in PBS and injected into 2-dpf-old larvae in ~10 nl via the caudal vein.

## Confocal microscopy and image-based quantification of infection

Larvae were embedded in 1.5% low melting point agarose (Thermo-Fisher) and a series of z stack images with a 2 µm step size was generated through the infected HBV. For infected THP-1 and A549 cells, a series of z stack images with a 1 µm step size was generated. Images were captured using the galvo scanner (laser scanner) of the Nikon A1R confocal microscope with a 20x Plan Apo 0.75 NA objective. Higher resolution images were generated using a 40x water-immersion Apo 1.15 NA objective. Bacterial burdens were determined by using the 3D surface-rendering feature of Imaris (Bitplane Scientific Software). Spread lipid images were generated by subtracting the bacterial surface from the lipid channel. 3D surface-rendering was then done in Imaris on both the total lipid and spread lipid images to generate a percent spread value (*Cambier et al., 2014b*).

## Macrophage and monocyte hindbrain recruitment assay

For the macrophage recruitment (*Figure 1G*): two dpf zebrafish were infected in the HBV with *M. marinum* at the dose reported in the figure legends. At 3 hr post infection, the number of total myeloid cells in the HBV was quantified using differential interference contrast microscopy using a 20x Plan Fluor 0.75 NA objective on Nikon's Ti eclipse inverted microscope. For quantification of monocytes (*Figure 5A and B*): two dpf zebrafish were injected in the caudal vein with 200 µg/ml of the nuclear stain Hoechst 33342 (Thermo-Fisher) 2 hr prior to HBV infection. Hoechst is unable to cross the blood-brain barrier and therefore will label circulating monocytes but will not label brain resident

macrophages (*Cambier et al., 2017*). At 3hpi blue-fluorescent cells in the hindbrain was quantified similar to above macrophage recruitment assay.

## Co-infection experiments

Around 40–50 tdTomato expressing wildtype *M. mairnum* were co-infected with an equal number of wasabi expressing *M. marinum* into the HBV. At three dpi the bacterial volume of the wildtype tdTomato expressing *M. marinum* was quantified as described above (*Confocal microscopy and image-based quantification of infection*) (*Cambier et al., 2014b*).

## Infectivity assay

3 days post-fertilization larvae were infected via the hindbrain ventricle with an average of 0.8 bacteria per injection. Fish harboring 1–3 bacteria were identified at 5 hr post infection by confocal microscopy. These infected fish were then evaluated at three dpi and were scored as infected or uninfected, based on the presence or absence of fluorescent bacteria (*Cambier et al., 2014b*).

## SDS-PAGE of lipid extracts

Petroleum ether extracts were dissolved by adding 30 µl of DMSO to 1 mg of lipids. 10 µl of 4x loading buffer (Licor) was then added to each sample and samples were heated at 95°C for 5 min. Samples were then separated on a 4–12% XT Bis-Tris Protein Gel (BioRad), in XT MES running buffer (BioRad) at 200 volts for 40 min. The gel was then stained with Colloidal Coomassie Brilliant Blue for 20 hr at room temperature with shaking. The gel was rinsed with water and then imaged.

## Fluorescence recovery after photobleaching (FRAP) experiments

FRAP experiments were performed using the galvo scanner (laser scanner) of the Nikon A1R confocal microscope with a 60x oil-immersion Plan Apo 1.4 NA objective. Photobleaching was performed with the 405 nm laser for 200 ms on a region of interest (ROI) encompassing ~1 µm from one pole of a single bacteria. A series of images was taken every second over the course of 31 s, one prior to bleaching. Labeled cells were mounted on a slide and coverslip in 0.75% low melting agarose. NIS Elements software (Nikon) was used to analyze the FRAP data to extract the fluorescence recovery kinetics. Briefly, the first image before photobleaching was used to generate an ROI for the entire cell and a second ROI was generated in the photobleached area. Total fluorescence intensities in both the whole cell area and the bleached area were extracted and normalized to correct for photobleaching of the dyes due to acquisition. The normalized fluorescence intensities of the bleached area were then fitted to a non-linear regression with a one-phase association, with the plateau values from each sample plotted to represent the mobile fraction (*Rodriguez-Rivera et al., 2017*).

## Exogenous lipid labeling of *M. marinum*

Wildtype or Δ*mmpL7 M. marinum* expressing wasabi fluorescent protein were grown in 7H9 medium supplemented with 10% OADC, 0.2% glycerol with or without 0.005% alkyne-cholesterol or azido-PE for 48 hr. Bacteria were then washed 3x with PBS prior to detection with fluorescent probes. For azido-PE, we followed the *Copper-free click chemistry of M. marinum* protocol above using DIBO-647. For alkyne-cholesterol we performed a copper-click reaction with AlexaFluor-647 Azide. For copper click: 400 µM BTTP (Click Chemistry Tools) and 200 µM copper sulfate (Sigma) were dissolved in PBS and allowed to complex for 20 min. 30 µM AlexaFluor-647 Azide (Thermo-Fisher) and 1.2 mM sodium ascorbate (Sigma) were then added to the solution. Bacterial pellets in 96-well v-bottom plates were resuspended in 50 µl of the solution and were incubated at 23°C protected from light for 45 min. Bacteria were then washed 5x in PBS prior to imaging on a Nikon A1R confocal microscope with a 60x oil-immersion Plan Apo 1.4 NA objective. Nikon elements software was used to determine fluorescent intensities of wasabi and cholesterol signals calculated from line profiles drawn perpendicular to bacterial membranes at least 0.5 µm from either pole.

## Cholesterol depletion and infection of A549 epithelial cells

A549 cells were seeded at 50,000 cells per 8-well Nunc Lab-Tek II chambered coverglass or at 100,000 cells per well in a 24 well plate. Cells were incubated at 37°C for 48 hr. Cells were washed 1x in PBS followed by treatment with 10 mM methyl-ß cyclodextrin (Sigma), 1 mM water-soluble

cholesterol (Sigma), or a combination of 10 mM methyl-ß cyclodextrin and 1 mM water-soluble cholesterol in serum free DMEM media. Cells were treated for 1 hr at 33°C followed by three washes with PBS. Cells plated on chambered coverglass were infected with azido-DIM labeled *M. marinum* at an MOI of 5 for 24 hr at 33°C followed by imaging on a Nikon A1R confocal microscope with a 20x Plan Apo 0.75 NA objective. 2 µm z-stacks were generated through the infected cells. Azido-DIM spreading was calculated similar to above (Section: *Confocal microscopy and image-based quantification of infection*). Cells plated on 24-well plates were rescued in DMEM + 10% FBS for 3 hr at 33°C. Cells were then harvested for quantification of cholesterol levels.

### Atorvastatin treatment of zebrafish larvae

At 48 hr post-fertilization, 0.5 µM atorvastatin (Sigma), or 10 µM water-soluble cholesterol (Sigma), or both were added to zebrafish water containing 1% DMSO. Control fish were incubated in water with 1% DMSO only. Zebrafish were incubated for 24 hr prior to infection or to cholesterol quantification. Drugs were replenished every 24 hr until experiment endpoint.

### Quantification of cholesterol in zebrafish and A549 epithelial cells

Cholesterol was quantified using the Total Cholesterol and Cholesterol Ester Colorimetric/Fluorometric Assay Kit (BioVision). For zebrafish, eight larvae were euthanized, transferred to a 1.5 ml Eppendorf tube and excess water was removed. A solution of chloroform:isopropanol:NP-40 (7:11:0.1) was added and the sample was sonicated in a water bath for 1 hr. Samples were then centrifuged at 16,000xg for 10 min and supernatants were transferred to a fresh tube and were allowed to dry in a 60°C water in a chemical fume hood. For A549 epithelial cells, chloroform:isopropanol: NP-40 (7:11:0.1) was added directly to cells in a 24 well plate. Cells were scrapped and solution was transferred to a 1.5 ml Eppendorf tube and was centrifuged at 16,000xg for 10 min and supernatants were transferred to a fresh tube and were allowed to dry in a 60°C water in a chemical fume hood. Dried lipids were then subjected to manufacturer's protocol and total cholesterol concentrations were determined by fluorescence on a SpectraMax i3x plate reader (Molecular Devices).

### Statistics

Statistical analyses were performed using Prism 8.4.3 (GraphPad): When appropriate D'Agostino-Pearson normality test was done to determine if all of the groups in a particular data set were of a gaussian distribution which then guided the subsequent statistical test performed (*Supplementary file 1*). Where the *n* value is given and not represented graphically in the figure, *n* represents the number of zebrafish used for each experimental group (*Figure 8H and I*).

## Acknowledgements

We thank Lalita Ramakrishnan for help interpreting data and manuscript writing, Karen Dobos for help purifying PDIM, Benjamin M Swarts for providing TreT-expressing *E. coli*, and David Tobin for zebrafish lines. This work was supported by a National Institutes of Health grant (AI51622) to CRB. CJC was supported by a Damon Runyon Postdoctoral Fellowship. SMB and JAB were supported by NIGMS F32 Postdoctoral Fellowships.

## Additional information

### Competing interests

Carolyn R Bertozzi: is a co-founder of OliLux Bio, Palleon Pharmaceuticals, InverVenn Bio, Enable Biosciences, and Lycia Therapeutics, and member of the Board of Directors of Eli Lilly. The other authors declare that no competing interests exist.

### Funding

| Funder | Grant reference number | Author |
| --- | --- | --- |
| National Institutes of Health | AI51622 | Carolyn R Bertozzi |

| Damon Runyon Cancer Research Foundation | Postdoctoral Fellowship | CJ Cambier |
|---|---|---|
| National Institutes of Health | F32 | Steven M Banik<br>Joseph A Buonomo |

The funders had no role in study design, data collection and interpretation, or the decision to submit the work for publication.

## Author contributions

CJ Cambier, Conceptualization, Investigation, Visualization, Methodology, Writing - original draft, Writing - review and editing; Steven M Banik, Conceptualization, Methodology, Writing - original draft, Writing - review and editing; Joseph A Buonomo, Investigation, Methodology, Writing - original draft, Writing - review and editing; Carolyn R Bertozzi, Conceptualization, Funding acquisition, Writing - original draft, Writing - review and editing

## Author ORCIDs

CJ Cambier (iD) https://orcid.org/0000-0002-0300-7377
Steven M Banik (iD) http://orcid.org/0000-0001-7070-3404
Carolyn R Bertozzi (iD) https://orcid.org/0000-0003-4482-2754

## Ethics

Animal experimentation: All experiments performed on zebrafish were in compliance with the U.S. National Institutes of Health guidelines. All animals were handled according to approved Stanford Institutional Animal Care and Use Committee protocol APLAC-30262.

## Decision letter and Author response

Decision letter https://doi.org/10.7554/eLife.60648.sa1
Author response https://doi.org/10.7554/eLife.60648.sa2

# Additional files

## Supplementary files

• Supplementary file 1. Summary of *P* values and statistical tests. Gaussian distribution was determined using the D'Agostino-Pearson normality test. The result of this test guided subsequent analyses.

• Supplementary file 2. NMR spectra.

• Transparent reporting form

## Data availability

All data generated or analyzed during this study are included in the manuscript and supporting files.

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
