## [Decision Letter]

**Acceptance summary:**

This manuscript builds on earlier work by this group investigating the role of phthiocerol dimycocerosate (PDIM) in the pathogenesis of *M. marinum* in the zebrafish. Herein, the authors describe an innovative chemical approach to track PDIM in vitro and in vivo. The authors find that bacterial PDIM spreads into host epithelial cell membranes to prevent immune activation. The authors show that host cholesterol promotes the spreading of PDIM in the membrane. This work establishes that interactions between host and pathogen lipids influence mycobacterial pathogenesis.

**Decision letter after peer review:**

Thank you for submitting your article "Spreading of a mycobacterial cell surface lipid into host epithelial membranes promotes infectivity" for consideration by *eLife*. Your article has been reviewed by Dominique Soldati-Favre as the Senior Editor, a Reviewing Editor, and three reviewers. The following individual involved in review of your submission has agreed to reveal their identity: Stefan H Oehlers (Reviewer #3).

The reviewers have discussed the reviews with one another and the Reviewing Editor has drafted this decision to help you prepare a revised submission.

Summary:

This manuscript builds on earlier work by this group investigating the role of phthiocerol dimycocerosate (PDIM) in the pathogenesis of *M. marinum* in the zebrafish. The authors reported that PDIM spreads into epithelial cells in order to inhibit TLR signaling and promote infectivity. The study describes an innovative chemical approach to track PDIM in vitro and in vivo, and presents several interesting results supporting their previous work and data from other groups. Major strengths are the authors' unique expertise and resources, especially in chemical modification of bacterial lipids and the zebrafish model of *M. marinum* infection. However, some of the data require additional controls and further experiments to conclusively support the authors' conclusion. In particular, (1) the conclusions regarding the PDIM/TLR axis require further dissection and consideration of alternative interpretations of the data presented, (2) the conclusions related to the spreading of PDIM into epithelial membranes require more controls to be convincing, and (3) the authors should compare their current results more extensively to previous findings to better illustrate the broader context of the topics addressed.

Essential revisions:

1) The evidence presented here suggests that TLRs might be inhibited by PDIM spreading in the membrane, however, the evidence provided here do not directly show this. It is possible that the cytokine response to δ mmpL7 mutants are distinct from WT *marinum* and this is the reason for the altered cellular recruitment that the authors observe. The authors suggest PDIM might disrupt TLR rafting into cholesterol rich domains and sensitivity of THP-1 cells to PDIM spreading. Could they combine their fluorescent PDIM tool with traditional TLR pathway analysis tools in THP-1 cells (where these tools presumably work better than intact zebrafish)? It seems likely that TLR dimerization or association with adaptors like MYD88 are likely mechanisms and should be straightforward to test. PDIM + vs – epithelial cells could also be isolated for proteomic or transcriptional analysis to provide some insight into the mechanism of PDIM inhibition of TLR sensitivity. Notably, the authors previous work suggested that PDIM masked TLR-ligands, do they observe the same masking effect with azido-DIM?

2) The microscopy is difficult to interpret in a convincing manner. While it is clear that there are DIM-488 foci that are not colocalized with the bacteria, it is not clear that this is in the host cell membrane. Colocalization with a host cell membrane dye would be a more convincing demonstration of this finding. Since it is not clear that the PDIM signal are in host membranes from the data provided, it raises the possibility that the spots are cytosolic lipid droplets. Could these foreign inclusions could be disrupting host cell biology? The proposal that PDIM interaction with cholesterol promotes spreading into epithelial membranes could be supported by labeling membrane cholesterol and examining how azido-DIM is distributed relative to cholesterol.

In addition, the effects of lipid reconstitution on the lipid and proteomic composition of the mycomembrane should be addressed and considered. How did chemical extraction and reconstitution help control the composition of the mycomembrane (subsection “Controlling Mycomembrane Composition by Chemical Extraction and Reconstitution”, Figure 1A-D)? Is the organization of added lipid similar to that of the lipids transported from the cytoplasm? Characterization of the envelope of mycobacteria reconstituted with azido-DIM relative to intact bacteria is important for interpreting the data on the spread of azido-DIM into the epithelial cell. Since some experiments last up to 3 days, what is the stability of the fluorescence signal of azido-DIM during bacterial growth and the time required for *M. marinum* to restore native PDIM in the envelope?

In Figure 1, the authors perform lipid-swap experiments with wild-type *M. marinum* and mmpL7 mutant, and MmpL7 transports PDIM and PGL in *M. marinum*, differences in PGL could also be contributing to the phenotypes.

In Figure 4, the authors propose that PDIM accumulated in "mycomembrane domains", but this term is poorly defined and images of the microdomains are not provided. The authors also state that mycomembrane domains containing PDIM have a uniquely high fluidity that would promote membrane mixing and contribute to PDIM spreading. Another possibility is that the membrane mixing may help PDIM to be transferred to the host membrane, where they could spread according to their diffusion properties. To test this, the authors could take advantage of A549 cells and compare the spread of DIM-499 in cells treated with DIM-499 or put in contact with DIM-499-labeled bacteria. Cell membranes are highly heterogeneous structures with distinct fluidity from one cell type to another. If PDIM can also diffuse in cell membranes, this could explain the difference in the kinetic of azido-DIM spreading between epithelial cells and macrophages.

For Figure 7, the rationale for the cholesterol study is interesting given that cholesterol is coupled to biosynthesis of methyl-branched lipids such as PDIM (for rev. Wilburn et al., 2018) and the role of DIM in macrophage invasion required cholesterol (Astarie-Dequeker et al., 2009). Figure 7A-D: Can the distinct hydrophobicity between *M. marinum* WT and mmpL7 mutant influence their capacity to sequester cholesterol. The authors could consider testing another hydrophobic lipid to further underscore this point.

3) Crucial data is also unnecessarily buried in supplementary figures. Supplemental Figure 3 is important to the interpretation of the results and needs to be more visible as part of a main figure. As there is no display item limit, so consider integrating the "validation" data into Figure 3 or splitting up Figure 3. Finally, the authors should compare their current results more extensively to previous findings to better illustrate the broader context of the topics addressed.

[Editors' note: further revisions were suggested prior to acceptance, as described below.]

Thank you for resubmitting your work entitled "Spreading of a mycobacterial cell surface lipid into host epithelial membranes promotes infectivity" for further consideration by *eLife*. Your revised article has been evaluated by Dominique Soldati-Favre (Senior Editor) and a Reviewing Editor.

The manuscript has been improved but there are some remaining issues that need to be addressed before acceptance, as outlined below:

The reviewers appreciate the careful attention of the authors to address most of the original comments, which has provided stronger support of their interpretations and conclusion. However, there are a few points that still need attention as detailed here. These comments should be able to be addressed by changes in the text.

1) Claims of TLR inhibition (starting in the abstract and throughout the manuscript) need to softened or removed as this is not directly addressed in the manuscript. The authors show effects on monocyte recruitment and that MyD88 deletion affects monocyte recruitment, but the experiments do not directly show PDIM is inhibiting TLRs and really just show that MyD88 is required for monocyte recruitment.

2) The authors should comment on the membrane effects of PDIM in the context of the literature (Augenstreich et al., 2019 and 2020). These recent data obtained with PDIM of *M. tuberculosis* would provide strong support for their findings.

3) Introduction: Could the authors clarify what fluidity means for a lipid? Membrane fluidity and lipid mobility may be more appropriate.

4) Subsection “Chemical Extraction and Reconstitution of the Mycomembrane of *M. marinum*”: The authors clearly showed that by mixing of extracted lipids with bacteria, the lipids coat *M. marinum*, but the authors do not demonstrate that the mycomembrane is reconstituted. For instance, what is the localization and orientation of lipids once coated on the bacteria in comparison to native lipids? This can be addressed by more specifically writing "coating of bacteria with extracted lipid" instead of "reconstitution of the mycomembrane".

5) subsection “Pre-infection PDIM Reservoirs are Required for Virulence”: The authors should mention and discuss that deletion of mas or mmpL7 also results in defects in PGL synthesis and localization.

6) Subsection “PDIM Spreads into 148 Host Macrophage Cell Membranes” and thereafter: The authors should define more clearly the notion of spreading. Did Azido-DIM spread within the same membrane, suggesting a lateral diffusion, and/or propagate to distinct membrane compartments?

7) subsection “PDIM’s Fluidity Promotes Spread into Epithelial Cell Membranes”: The authors claimed that PDIM's fluidity correlates with the lipid's ability to spread into host membrane. Another possibility could be that PDIM's ability to diffuse into the bacterial membrane increases the membrane fluidity, which facilitates membrane mixing. The consequence would be the transfer of PDIM to the host membrane, in which the lipids inserted and can diffuse laterally, as previously reported (Augenstreich et al., 2019, 2020). The effect of PDIM on bacterial membrane fluidity could therefore be an explanation for the transfer of the lipid rather than for its spreading. In contrast, TMMs are less capable of diffusing and getting transferred. TMM are amphiphilic molecules in comparison to the highly hydrophobic PDIM. This strongly suggests a relationship between lipid structure and capacity to get transferred to host membranes. Please discuss this point in the Discussion section.

8) The authors should be clear in their conclusions and the Discussion that the data have been obtained with bacteria coated with extracted lipids.

9) The additional images provided by the authors are encouraging but there is not sufficient detail about the quantification to understand if this is representative or not. Furthermore, while the authors are showing overlapping signals with the plasma membrane, these images do not prove that PDIM is in the membrane. To show that PDIM is in the membrane, a z stack is required but not provided.

---

## [Author Response]

Essential revisions:1) The evidence presented here suggests that TLRs might be inhibited by PDIM spreading in the membrane, however, the evidence provided here do not directly show this. It is possible that the cytokine response to δ mmpL7 mutants are distinct from WT marinum and this is the reason for the altered cellular recruitment that the authors observe.

We absolutely agree with the reviewers that different cytokine responses are most likely underlying the different cell recruitment phenotypes seen toward wildtype and ∆*mmpL7 M. marinum*. From prior work (Cambier et al., 2013), we know that wildtype bacteria induce the chemokine CCL2 and that expression of the receptor CCR2 on monocytes is required for their ability to be recruited to the hindbrain ventricle. In this setting, TLR-signaling is not required for monocyte recruitment. Alternatively, when bacteria lack PDIM, we know that CCR2 signaling is no longer required to recruit monocytes but now TLR signaling is required. What we didn’t know at the time was when bacteria lack PDIM, where is detection by TLRs occurring. In this study our data suggests that TLR signaling is happening immediately following infection at epithelial cells at the site of infection. Presumably this then leads to the induction of cytokines/chemokines that are distinct from what occurs during wildtype bacterial infection. Identification of which cytokines are functionally responsible for this phenotype is ongoing.

The authors suggest PDIM might disrupt TLR rafting into cholesterol rich domains and sensitivity of THP-1 cells to PDIM spreading. Could they combine their fluorescent PDIM tool with traditional TLR pathway analysis tools in THP-1 cells (where these tools presumably work better than intact zebrafish)? It seems likely that TLR dimerization or association with adaptors like MYD88 are likely mechanisms and should be straightforward to test. PDIM + vs – epithelial cells could also be isolated for proteomic or transcriptional analysis to provide some insight into the mechanism of PDIM inhibition of TLR sensitivity. Notably, the authors previous work suggested that PDIM masked TLR-ligands, do they observe the same masking effect with azido-DIM?

We are currently searching for the most appropriate in-vitro model system to address this hypothesis. Our data in this manuscript demonstrates that PDIM is required to spread into epithelial cells to prevent recruitment of microbicidal monocytes. Because of this, we don’t anticipate data generated in monocyte cell lines to be relevant. Furthermore, previous literature suggests that TLR signaling may not be fully functional in A549 epithelial cells (DOI: 10.1371/journal.pone.0021827). Nevertheless, we evaluated the ability of A549 epithelial cells to respond to TNFa, LPS (TLR4 agonist), and PAM3CSK4 (TLR2 agonist) using an NFkB luciferase reporter A549 cell line (BPS Bioscience, Cat# 60625). Following an overnight incubation, the cells clearly respond to TNFa stimulation but do not respond to TLR agonists (See data in Author response image 1). In light of these data we are pursuing more relevant in vitro epithelial model systems where TLR signaling is intact to evaluate how PDIM disrupts this response. With regards to masking TLR ligands, our current data cannot rule out this mechanism, however it suggests that PDIM plays a more active role through the occupation of host membranes to silence TLR responses (See Discussion).

**Author response image 1. sa2fig1:** 

2) The microscopy is difficult to interpret in a convincing manner. While it is clear that there are DIM-488 foci that are not colocalized with the bacteria, it is not clear that this is in the host cell membrane. Colocalization with a host cell membrane dye would be a more convincing demonstration of this finding. Since it is not clear that the PDIM signal are in host membranes from the data provided, it raises the possibility that the spots are cytosolic lipid droplets. Could these foreign inclusions could be disrupting host cell biology? The proposal that PDIM interaction with cholesterol promotes spreading into epithelial membranes could be supported by labeling membrane cholesterol and examining how azido-DIM is distributed relative to cholesterol.

We appreciate the reviewers’ scrutiny with regards to our microscopy data. We have included additional experiments that support our claims that PDIM is spreading into non-macrophage cell membranes. First, we took advantage of the transgenic zebrafish line Tg(flk1:mcherry), whose vascular endothelial cells express the fluorescent protein mcherry. Following intravenous infection with DIM-488 labeled mycobacteria we captured bacteria in contact with endothelial cells with spread DIM-488 co-localizing with the endothelial marker (Figure 4B, subsection “PDIM’s Fluidity Promotes Spread into Epithelial Cell Membranes”). Additionally, we used the plasma membrane label Alexa-fluor 594 wheat germ agglutinin to label A549 epithelial cells. We then captured DIM-488 having spread and co-localizing with the plasma membrane label following infection (Figure 4C, subsection “P_dim_’s Fluidity Promotes Spread into Epithelial Cell Membranes”). Finally, we labeled A549 epithelial cells with BODIPY-cholesterol and imaged PDIM spreading onto these cells and found PDIM to strongly co-localize with BODIPY-cholesterol (Figure 8C, Discussion section).

In addition, the effects of lipid reconstitution on the lipid and proteomic composition of the mycomembrane should be addressed and considered. How did chemical extraction and reconstitution help control the composition of the mycomembrane (subsection “Controlling Mycomembrane Composition by Chemical Extraction and Reconstitution”, Figure 1A-D)? Is the organization of added lipid similar to that of the lipids transported from the cytoplasm? Characterization of the envelope of mycobacteria reconstituted with azido-DIM relative to intact bacteria is important for interpreting the data on the spread of azido-DIM into the epithelial cell. Since some experiments last up to 3 days, what is the stability of the fluorescence signal of azido-DIM during bacterial growth and the time required for M. marinum to restore native PDIM in the envelope?

The chemical extraction and reconstitution method allows us to control the lipid content of the outer mycomembrane, as is evidenced by later sections in the paper. As such we have renamed this subsection “Chemical Extraction and Reconstitution of the Mycomembrane of *M. marinum*”. To address concerns over the lipid content of the mycomembrane pre and post reconstitution we extracted lipids from untreated or reconstituted bacteria and compared them by TLC. No apparent differences were noticed (Figure 1—figure supplement 1D, subsection “Chemical Extraction and Reconstitution of the Mycomembrane of *M. marinum*”). We additionally ran proton NMRs of the two crude extracts from control or reconstituted bacteria, the spectra were near identical (Materials and method subsection “NMR spectra”). To evaluate how extraction may alter proteins on the mycomembrane we first asked if any proteins were removed during petroleum ether extraction. We resuspended 1 mg of lipid extract in 30µl of DMSO and separated the sample by SDS-PAGE followed by staining overnight with Colloidal Coomassie Brilliant Blue. Similar to previous findings (Moliva et al., 2019) we found no evidence of protein in the petroleum ether extracts (Figure 1—figure supplement 1E, subsection “Chemical Extraction and Reconstitution of the Mycomembrane of *M. marinum*”), thus we do not anticipate any changes to the mycomembrane protein composition arising following reconstitution. Furthermore, the ability of reconstituted bacteria to survive as well as untreated bacteria following infection further supports this idea. Additionally, we evaluated how long it takes for delipidated bacteria to repopulate their mycomembrane lipids. Immediately following extraction, delipidated bacteria were rescued into complete growth medium. Every 24hrs these bacteria were harvested and re-extracted and the amount of lipids removed was measured. Even after 5 days in culture, bacteria did not fully restore their mycomembrane lipids (Figure 1—figure supplement 1B, subsection “Chemical Extraction and Reconstitution of the Mycomembrane of *M. marinum*”). Finally, we grew DIM-488 labeled bacteria for several days following reconstitution and found that the intensity of DIM-488 decreased but was still present on bacteria even after 3 days in culture (Figure 3D and E, subsection “P_dim_ Spreads into Host Macrophage Cell Membranes in vivo”), suggesting that PDIM is not appreciably shed during mycobacterial growth in aqueous medium and that PDIM spreading is a result of interactions with the host.

In Figure 1, the authors perform lipid-swap experiments with wild-type M. marinum and mmpL7 mutant, and MmpL7 transports PDIM and PGL in M. marinum, differences in PGL could also be contributing to the phenotypes.

The reviewers are correct that phenolic glycolipid (PGL) also depends on MmpL7 for its localization to the mycomembrane. Using mutants defective for PGL synthesis, that retain PDIM on their surface, previous studies found that these phenotypes (decreased growth following hindbrain infection with ~100 bacteria and TLR-dependent monocyte recruitment following hindbrain infection with ~100 bacteria) were not influenced by the absence of PGL (Cambier et al., 2013). Therefore, we decided to leave this detail of mycobacterial cell wall biology out of the manuscript so as not to overwhelm the readers.

In Figure 4, the authors propose that PDIM accumulated in "mycomembrane domains", but this term is poorly defined and images of the microdomains are not provided. The authors also state that mycomembrane domains containing PDIM have a uniquely high fluidity that would promote membrane mixing and contribute to PDIM spreading. Another possibility is that the membrane mixing may help PDIM to be transferred to the host membrane, where they could spread according to their diffusion properties. To test this, the authors could take advantage of A549 cells and compare the spread of DIM-499 in cells treated with DIM-499 or put in contact with DIM-499-labeled bacteria. Cell membranes are highly heterogeneous structures with distinct fluidity from one cell type to another. If PDIM can also diffuse in cell membranes, this could explain the difference in the kinetic of azido-DIM spreading between epithelial cells and macrophages.

We agree with the reviewers that our interpretation that PDIM resides in distinct membrane domains is not accurate. We have changed our description to better reflect our data. We also appreciate your inquiry into the mechanism behind PDIM spreading. Your suggestion that membrane mixing may help PDIM transfer to host epithelial membranes is interesting. We agree that membrane mixing is most likely an important step for PDIM transfer, however, if mixing were the deterministic mechanism driving lipid transfer then we would expect trehalose monomycolate to also be able to spread into epithelial cells, which it does not. Additionally, our data demonstrating that methyl-branched mycocerosic acids promote PDIM fluidity and are required for spreading into epithelial membranes supports the hypothesis that the biophysical properties of PDIM on the bacterial surface promote its ability to spread into host membranes.

Regarding the different kinetics of spreading between epithelial cells and macrophages, we do not believe our data suggest that the rate of spreading is different between the two. In fact, our data cannot speak to the rate of lipid spread into host membranes. Our data shows that PDIM spreads into epithelial membranes before it spreads into macrophage membranes. We believe this is a result of the physiology of infection. In our infection model, mycobacteria remain in an extracellular space for around 3-5 hours following infection after which they are phagocytosed by macrophages. While in this extracellular space they can make contact with epithelial cells. Thus, PDIM spreads into epithelial membranes prior to macrophage membranes because bacteria make contact with epithelial cells prior to interacting with/being phagocytosed by macrophages. A better explanation of these kinetics are now in subsection “P_dim_’s Fluidity Promotes Spread into Epithelial Cell Membranes”.

For Figure 7, the rationale for the cholesterol study is interesting given that cholesterol is coupled to biosynthesis of methyl-branched lipids such as PDIM (for rev. Wilburn et al., 2018) and the role of DIM in macrophage invasion required cholesterol (Astarie-Dequeker et al., 2009).

We thank the reviewers for highlighting these important studies. We agree that the Wilburn review highlights an important link between cholesterol metabolism and PDIM bio-synthesis. We have included this in our Discussion. For the introduction to Figure 7 we were motivated by studies that suggested a physical association between PDIM and cholesterol outside of macrophages, as this suggested to us that cholesterol may be promoting the spreading into epithelial cells that we were observing.

The work from Astarie-Dequeker et al., does more directly demonstrate an interaction between PDIM and cholesterol. Indeed, insubsection “P_dim_ Spreads into Epithelial Membranes” we discuss these results. However, in contrast to Astarie-Dequeker et al., we found no difference in the rate of phagocytosis of wildtype or PDIM-deficient *M. marinum* following infection of zebrafish larvae (Figure 3H). Therefore, we decided not to further describe the mechanism from Astarie-Dequeker et al., where they show cholesterol being required for PDIM-promoted phagocytosis as we felt it may not explain the results in our model.

Figure 7A-D: Can the distinct hydrophobicity between M. marinum WT and mmpL7 mutant influence their capacity to sequester cholesterol. The authors could consider testing another hydrophobic lipid to further underscore this point.

The reviewers raise an excellent point. The ability of mycobacteria to sequester cholesterol could be due to their hydrophobic outer cell membrane and not due to interactions with PDIM directly. To test this idea we chose a compound that was just as hydrophobic as cholesterol but structurally very distinct. We tested whether wildtype or ∆mmpL7 *M. marinum* could sequester 16:0 azidocaproyl phosphoethanolamine (azido-PE, Figure 8—figure supplement 1B). We found that either strain were capable of sequestering this hydrophobic compound (Figure 8—figure supplement 1C and D, Discussion), suggesting that there is not dramatic differences in the hydrophobicity of wildtype and ∆mmpL7 *M. marinum*.

3) Crucial data is also unnecessarily buried in supplementary figures. Supplemental Figure 3 is important to the interpretation of the results and needs to be more visible as part of a main figure. As there is no display item limit, so consider integrating the "validation" data into Figure 3 or splitting up Figure 3. Finally, the authors should compare their current results more extensively to previous findings to better illustrate the broader context of the topics addressed.

We appreciate that the reviewers consider our supplemental data important to the understanding of our manuscript. As such we have taken their advice and have added these panels to the main figures and have split figure three into two figures.

[Editors' note: further revisions were suggested prior to acceptance, as described below.]

The manuscript has been improved but there are some remaining issues that need to be addressed before acceptance, as outlined below:The reviewers appreciate the careful attention of the authors to address most of the original comments, which has provided stronger support of their interpretations and conclusion. However, there are a few points that still need attention as detailed here. These comments should be able to be addressed by changes in the text.1) Claims of TLR inhibition (starting in the abstract and throughout the manuscript) need to softened or removed as this is not directly addressed in the manuscript. The authors show effects on monocyte recruitment and that MyD88 deletion affects monocyte recruitment, but the experiments do not directly show PDIM is inhibiting TLRs and really just show that MyD88 is required for monocyte recruitment.

The claims of TLR inhibition have been softened and instead are more focused on Myd88 signaling and the resulting recruitment of microbicidal monocytes. These changes were made throughout the manuscript.

2) The authors should comment on the membrane effects of PDIM in the context of the literature (Augenstreich et al., 2019 and 2020). These recent data obtained with PDIM of M. tuberculosis would provide strong support for their findings.

We have discussed these works in the Discussion.

3) Introduction: Could the authors clarify what fluidity means for a lipid? Membrane fluidity and lipid mobility may be more appropriate.

You are correct and this language has been corrected throughout the manuscript.

4) Subsection “Chemical Extraction and Reconstitution of the Mycomembrane of M. marinumof the Mycomembrane of M. marinum”: The authors clearly showed that by mixing of extracted lipids with bacteria, the lipids coat M. marinum, but the authors do not demonstrate that the mycomembrane is reconstituted. For instance, what is the localization and orientation of lipids once coated on the bacteria in comparison to native lipids? This can be addressed by more specifically writing "coating of bacteria with extracted lipid" instead of "reconstitution of the mycomembrane".

Reconstitution has been replaced with recoating throughout the manuscript.

5) subsection “Pre-infection PDIM Reservoirs are Required for Virulence”: The authors should mention and discuss that deletion of mas or mmpL7 also results in defects in PGL synthesis and localization.

We have brought this to the reader’s attention (subsection “PDIM Spreads into Macrophage Membranes”).

6) Subsection “PDIM Spreads into 148 Host Macrophage Cell Membranes” and thereafter: The authors should define more clearly the notion of spreading. Did Azido-DIM spread within the same membrane, suggesting a lateral diffusion, and/or propagate to distinct membrane compartments?

Our data suggests both to be taking place. We see spreading appearing next to where bacteria make contact with host membranes as well as intracellular punctate forming. We have made clearer these observations (subsection “PDIM’s Mobility Promotes Spread into Epithelial Cell Membranes”).

7) subsection “PDIM’s Fluidity Promotes Spread into Epithelial Cell Membranes”: The authors claimed that PDIM's fluidity correlates with the lipid's ability to spread into host membrane. Another possibility could be that PDIM's ability to diffuse into the bacterial membrane increases the membrane fluidity, which facilitates membrane mixing. The consequence would be the transfer of PDIM to the host membrane, in which the lipids inserted and can diffuse laterally, as previously reported (Augenstreich et al., 2019, 2020). The effect of PDIM on bacterial membrane fluidity could therefore be an explanation for the transfer of the lipid rather than for its spreading. In contrast, TMMs are less capable of diffusing and getting transferred. TMM are amphiphilic molecules in comparison to the highly hydrophobic PDIM. This strongly suggests a relationship between lipid structure and capacity to get transferred to host membranes. Please discuss this point in the Discussion section.

We have expanded on how our data speaks to these more basic principles of lipid dynamics (Discussion).

8) The authors should be clear in their conclusions and the Discussion that the data have been obtained with bacteria coated with extracted lipids.

We make have made this clear in the Discussion.

9) The additional images provided by the authors are encouraging but there is not sufficient detail about the quantification to understand if this is representative or not. Furthermore, while the authors are showing overlapping signals with the plasma membrane, these images do not prove that PDIM is in the membrane. To show that PDIM is in the membrane, a z stack is required but not provided.

We have included a z projection of DIM-488 spreading into the plasma membrane of A549 cells. (Figure 4C).